



# Data-based estimates of interannual sea–air $CO_2$ flux variations 1957–2020 and their relation to environmental drivers

Christian Rödenbeck[1], Tim DeVries[2], Judith Hauck[3], Corinne Le Quéré[4], and Ralph F. Keeling[5]

[1]Max Planck Institute for Biogeochemistry, Jena, Germany
[2]Department of Geography, University of California, Santa Barbara, USA
[3]Alfred Wegener Institute Helmholtz Centre for Polar and Marine Research, Bremerhaven, Germany
[4]School of Environmental Sciences, University of East Anglia, Norwich, UK
[5]Scripps Institution of Oceanography, University of California, San Diego, USA

*Correspondence to:* C. Rödenbeck (christian.roedenbeck@bgc-jena.mpg.de)

**Abstract.** This study considers year-to-year and decadal variations as well as secular trends of the sea–air $CO_2$ flux over the 1957–2020 period, as constrained by the $pCO_2$ measurements from the SOCATv2021 data base. In a first step, we relate interannual anomalies in ocean-internal carbon sources and sinks to local interannual anomalies in sea surface temperature (SST), the temporal changes of SST (dSST/dt), and squared wind speed ($u^2$), employing a multi-linear regression. In the
tropical Pacific, we find interannual variability to be dominated by dSST/dt, as arising from variations in the upwelling of colder and more carbon-rich waters into the mixed layer. In the eastern upwelling zones as well as in circumpolar bands in the high latitudes of both hemispheres, we find sensitivity to wind speed, compatible with the entrainment of carbon-rich water during wind-driven deepening of the mixed layer and wind-driven upwelling. In the Southern Ocean, the secular increase in wind speed leads to a secular increase in the carbon source into the mixed layer, with an estimated reduction of the sink trend
in the range 17 to 42%. In a second step, we combined the result of the multi-linear regression and an explicitly interannual $pCO_2$-based additive correction into a "hybrid" estimate of the sea–air $CO_2$ flux over the period 1957–2020. As a $pCO_2$ mapping method, it combines (a) the ability of a regression to bridge data gaps and extrapolate into the early decades almost void of $pCO_2$ data based on process-related observables and (b) the ability of an autoregressive interpolation to follow signals even if not represented in the chosen set of explanatory variables. The "hybrid" estimate can be applied as ocean flux prior for
atmospheric $CO_2$ inversions covering the whole period of atmospheric $CO_2$ data since 1957.

## 1 Introduction

The atmospheric $CO_2$ content has risen during the recent decades, primarily due to anthropogenic emissions (IPCC, 2013). However, the actual rise has been co-determined by the exchange of $CO_2$ between the atmosphere and natural systems, notably the ocean and the land vegetation. The uptake of atmospheric $CO_2$ into the ocean is primarily driven by the solution
disequilibrium across the sea–air interface. As the surface-ocean carbon content is lagging behind the atmospheric rise, the ocean uptake is, to first order, increasing in parallel with the atmospheric $CO_2$ rise. However, natural climate variability and anthopogenic climate change alter the uptake rate on year-to-year and decade-to-decade time scales as well as in its secular





evolution. This leads to a feedback loop: Atmospheric $CO_2$ influences the climate via the greenhouse effect, while the climate in turn influences the carbon-relevant natural systems in the ocean and on land. This feedback loop can dampen or accelerate climate change.

In order to understand the future climate trajectory, therefore, we need to quantitatively understand the carbon response of
the natural systems. For example, how will secular trends towards higher wind speeds in the Southern Ocean affect the sea–air $CO_2$ exchange in this region (Le Quéré et al., 2007; Hauck et al., 2013, and many others)? While the relevant time scale is secular (multi-decadal) trends, year-to-year or decade-to-decade variability in $CO_2$ fluxes can be used as "natural experiments" to understand the climatic controls of the land and ocean carbon cycle. This can be done by quantifying variations of carbon fluxes from suitable observations, and statistically relating them to variations in quantities describing relevant environmental
conditions. Even though the climate–carbon cycle feedback loop involves the global $CO_2$ fluxes only (because atmospheric $CO_2$ is mixed globally within about one year), the statistical analysis needs to be done on a spatial scale fine enough to accomodate the spatial inhomogeneity of the involved processes.

Suitable observational data therefore need to provide sufficient spatial and temporal detail and span several decades. Regarding ocean $CO_2$ fluxes, there are essentially two types of such data: (1) Sustained *atmospheric $CO_2$ measurements* at various
locations worldwide (Keeling, 1978; Conway et al., 1994; Francey et al., 2003, and many more) and (2) sustained and spatially extensive measurements of the $CO_2$ *partial pressure ($pCO_2$) in the surface ocean* (Bakker et al., 2016). As changes and gradients in atmospheric $CO_2$ reflect the sum of the regional $CO_2$ sources and sinks at the surface, atmospheric $CO_2$ data have been combined with simulations of atmospheric tracer transport and inverse techniques to estimate spatial and temporal variations of the $CO_2$ fluxes ("atmospheric inversion", Newsam and Enting, 1988; Rayner et al., 1999; Bousquet et al., 2000;
Rödenbeck et al., 2003; Baker et al., 2006, and many others). Even though most of the atmospheric inversions start in the 1990s or 2000s when more and more stations became operational, the longest time series of atmospheric $CO_2$ measurements are available since 1957 (as used in Rödenbeck et al., 2018a). However, atmospheric inversions are known to have limited capability to correctly assign signals to land or ocean (Peylin et al., 2013). While the resulting error is relatively small for the land fluxes, it strongly affects the estimated ocean flux variability, because the ocean variability is much smaller than the land
variability.

Therefore, the surface-ocean $pCO_2$ data (Bakker et al., 2016) currently provide the most detailed information about the spatio-temporal variability of the sea–air $CO_2$ exchange. To cope with the very inhomogeneous distribution of these $pCO_2$ data in space and time, including substantial gaps, several methods have been developed to map (interpolate) the data into continuous spatio-temporal fields of $pCO_2$ (Takahashi et al., 2009; Watson et al., 2009; Valsala and Maksyutov, 2010; Landschützer et al.,
2013; Nakaoka et al., 2013; Rödenbeck et al., 2013; Majkut et al., 2014; Iida et al., 2015; Jones et al., 2015; Zeng et al., 2015; Denvil-Sommer et al., 2019; Gregor et al., 2019, and several others). Most of these mappings employ either (i) an auto-regressive interpolation that fills unobserved areas or periods based on the neighbouring data within some prescribed correlation radii in space and time, or (ii) a regression of $pCO_2$ against suitable explanatory variables that have been observed more densely and over all the target period (using linear regression, neural networks, or machine learning). These two types
of mappings offer complementary advantages, as regressions against explanatory variables possess predictive skill allowing



to fill longer data gaps (and potentially to extrapolate into data-void periods), while auto-regressive mappings can reproduce all signals in the data even if not represented in the chosen explanatory variables (Rödenbeck et al., 2015). From the mapped $pCO_2$ fields, the sea–air $CO_2$ flux is then calculated via a gas-exchange parameterization. In addition to studying the ocean carbon cycle, these flux estimates have also been used as an interannually varying ocean prior in atmospheric $CO_2$ inversions

to potentially improve land $CO_2$ flux estimates (Rödenbeck et al., 2014).

With regard to the aim of understanding how the oceanic carbon cycle may respond to decadal and secular climatic changes as laid out above, however, the current $pCO_2$ mappings have two limitations. As a first limitation, the current $pCO_2$ mappings only provide spatio-temporal variations in the $pCO_2$ field and the sea-air $CO_2$ flux, but do not explicitly quantify the relationships between these variations and underlying environmental drivers. This is true even for the regressions against

explanatory variables: Even though these relationships are implicitly contained in the synaptic weights of neural networks or similar parameters in machine learning algorithms, they are not accessible from these algorithms in interpretable form.

The second limitation arises from the fact that very little $pCO_2$ data exist before the mid 1980s (Bakker et al., 2016). In the equatorial Pacific, critical due to its large variability, sufficient coverage does not start before 1992. Despite their predictive skill, even the available $pCO_2$ regressions against explanatory variables only cover a time period not longer or even shorter

than the $pCO_2$ data period, some for example because chlorophyll-a data have only been available in the satellite era since 1997. Thus, none of the currently available $pCO_2$ mappings starts before 1980. Consequently, they cannot be used as a data-based ocean prior in atmospheric $CO_2$ inversions over the full period of atmospheric data (1957–present). Further, the $pCO_2$ mappings do not cover the 1960–present period considered in ongoing synthesis projects like the annual carbon budget by the Global Carbon Project (GCP) (Friedlingstein et al., 2020), which so far exclusively relies on process model simulations during

the first decades.

In this study, we aim to contribute to overcome these two limitations. First, extending the CarboScope $pCO_2$ mapping (Rödenbeck et al., 2013, 2014), we have developed a multi-linear regression explicitly estimating the sensitivities of the carbon sources and sinks in the oceanic mixed layer against the variations in relevant explanatory variables. Second, we have combined this multi-linear regression with an additive auto-regressive correction into a "hybrid" mapping, inheriting the complementary

advantages of both auto-regressive and regression-based $pCO_2$ mappings. As the regression extrapolates the variability back to 1957 by only using explanatory variables available throughout all this time frame, the hybrid mapping yields an observation-based estimate of the spatio-temporal variability of sea–air $CO_2$ fluxes since 1957.

After describing the mapping methods (Sect. 2), we present the resulting interannual variations of sea–air $CO_2$ fluxes (Sects. 3.1–3.2). We evaluate the predictive skill of the multi-linear regression step as one of its most important requirements

(Sect. 3.3). We show the contributions of the individual environmental variables to the interannual $CO_2$ flux variations (Sect. 3.4). Finally, we present spatial patterns in the regression coefficients (sensitivities) and discuss possible underlying mechanisms controlling sea–air $CO_2$ fluxes (Sect. 3.5). In the discussion, we consider whether the presented multi-linear regression indeed meaningfully reflects biogeochemical processes (Sect. 4.1), which fraction of interannual variability it is able to capture (Sect. 4.2), to which extend the sensitivities depend on time-scale (Sect. 4.3), and how some uncertainties may

affect the result (Sects. 4.4–4.5). In an appendix, we focus on the global total sea–air $CO_2$ flux estimated by the hybrid mapping



in terms of its mean (Sect. A1) and secular trend (Sect. A2), discussing its uncertainty and comparing it with literature values obtained by other methods.

## 2 Method

### 2.1 $p\mathrm{CO_2}$ mapping

5 #### 2.1.1 Overview

The $p\mathrm{CO_2}$ mapping schemes used in this study are variants of the CarboScope $p\mathrm{CO_2}$ mapping described in Rödenbeck et al. (2013). The estimates are based on the $p\mathrm{CO_2}$ data in the SOCAT data collection version v2021 (Bakker et al., 2016, 2020). The elements common to all mapping variants are summarized in the following and illustrated in Fig. 1; for details we refer to Rödenbeck et al. (2013).

10     Parameterizations of sea–air gas exchange (quadratic wind speed dependence as in Wanninkhof, 1992) and solubility (Weiss, 1974), a calculation of the chemical equilibrium of the carbonate chemistry in sea water (Orr and Epitalon, 2015), as well as a mixed-layer budget of dissolved inorganic carbon (DIC) (Rödenbeck et al., 2013), are used to express the $p\mathrm{CO_2}$ field and the sea–air $\mathrm{CO_2}$ flux field as a function of the ocean-internal flux of DIC, $f_{\mathrm{int}}$ (Fig. 1). The ocean-internal DIC flux $f_{\mathrm{int}}$ is meant to comprise all sources and sinks of DIC into or out of the oceanic mixed layer, through biological conversion within the mixed 15 layer or through mixing-in of waters with different DIC concentration. It is expressed as the sum of a fixed (a-priori) flux field and a set of predefined spatio-temporal patterns of adjustment each scaled by an adjustable parameter (the sets of patterns are detailed for each variant of the mapping below). Then, the mismatch between the calculated $p\mathrm{CO_2}$ field (at the respective pixels and time steps containing the SOCAT $p\mathrm{CO_2}$ samplings) and the corresponding measured $p\mathrm{CO_2}$ values (black dots in the $p\mathrm{CO_2}$ panel of Fig. 1) is gauged by a quadratic cost function. The (a-posteriori) estimates of the mapping are calculated 20 from those values of the adjustable parameters that minimize this cost function. In the example of Fig. 1, the two estimates (colored) follow the data points (black dots) more closely than the prior (grey).

    Spatial and temporal interpolation between the very inhomogeneously sampled data is implemented in the following way. By choosing a set of spatial patterns of adjustment that are centered at all the individual ocean pixels but simultaneously affect the respective neighbouring pixels within some correlation radius (to be detailed below), in conjunction with additional 25 Bayesian terms in the cost function that penalize large adjustments to the adjustable parameters, the ocean-internal DIC flux field is forced to be smooth. These smoothness constraints spread the information from data-covered pixels to neighbouring unconstrained pixels (see Fig. 5 of Rödenbeck et al., 2013), thereby interpolating spatial data gaps. (The set of patterns of adjustment indirectly defines the Bayesian a-priori covariance matrix, see Rödenbeck, 2005, for background.) Interpolation in time is achieved analoguously by temporal smoothness constraints (even though, for practical reasons, a mathematically 30 equivalent Fourier formulation is used).

    The four mapping variants used here differ in the choices of the prior for $f_{\mathrm{int}}$ and the set of spatio-temporal patterns of adjustment. Our development started from a variant (Sect. 2.1.2) essentially identical to Rödenbeck et al. (2013) used as the



CarboScope $pCO_2$ mapping before version v2020, except for some technical changes described later (Sect. 2.1.6–2.1.7). As an intermediate modification, we introduced a prior stabilizing the secular trend (Sect. 2.1.3); the result of this variant will be used to help discussing specific aspects. The main results of this study come from the multi-linear regression (Sect. 2.1.4) and the hybrid mapping (Sect. 2.1.5). Fig. 2 summarizes the differences and the flow of information between the four variants.

### 2.1.2 The "Zero-prior Explicitly interannual" $pCO_2$ mapping (ZE)

The starting variant has a general set of (many) patterns of adjustment, allowing an arbitrary smooth spatio-temporal internal DIC flux field $f_{int}^{ZE}$ (Rödenbeck et al., 2013). This field $f_{int}^{ZE}$ is implemented as the sum of a constant term (subscript 'LT' for 'long-term') and terms for seasonal (subscript 'Seas') and interannual anomalies (non-seasonal, subscript 'IAV'),

$$
\begin{aligned}
f_{int}^{ZE}(x,y,t) &= f_{int,IAV}^{adj}(x,y,t) \\
&+ f_{int,LT}^{ADJ}(x,y) + f_{int,Seas}^{ADJ}(x,y,s)
\end{aligned}
\tag{1}
$$

As indicated by the superscript 'adj' or 'ADJ' (difference explained below), all these terms involve degrees of freedom being adjusted in the cost function minimization sketched above. A-priori, all adjustable terms are zero, such that the prior of $f_{int}^{ZE}$ is zero as well.

The interannual term $f_{int,IAV}^{adj}(x,y,t)$ can represent non-seasonal anomalies on all month-to-month, year-to-year, or decadal time scales, including secular trends. The level of its temporal smoothness corresponds to a-priori correlation length scales of about 4 weeks, implemented through a mathematically equivalent Fourier series with dampened higher-frequency components (where Fourier terms dampened to less than 2% are discarded entirely). This amounts to 722 scalable Fourier terms for our 71-year calculation period 1951–2021. The seasonal term $f_{int,Seas}^{ADJ}$ only contains seasonal Fourier components; thus it only depends on the time $s$ within the year and repeats itself every year. Along the seasonal cycle, it has the same temporal correlation length as the interannual term of about 4 weeks, amounting to 10 scalable Fourier terms. The constant term $f_{int,LT}^{ADJ}$ is not time-dependent by definition (1 temporal degree of freedom).

Spatially, the level of smoothness in all three terms corresponds to a-priori correlation length scales of about $640\,km$ in longitude and latitude.

As symbolized by the capitalized superscript 'ADJ', the relative a-priori uncertainties of $f_{int,LT}^{ADJ}$ and $f_{int,Seas}^{ADJ}$ at any time step are chosen to be larger than those of $f_{int,IAV}^{adj}$, corresponding to larger expected amplitudes of seasonal Fourier components of $f_{int}$ compared to non-seasonal ones. In terms of the implied a-priori autocorrelation function, these enhanced a-priori uncertainties of seasonal variations are equivalent to non-zero temporal correlations between the flux at any given time-of-year and the same time-of-year in all other years (in addition to the 4-weeks decaying correlations mentioned above). Due to these periodic correlations, $f_{int}$ in time periods without data does not fall back to the prior (here zero) but to the mean seasonal cycle $f_{int,Seas}^{ADJ}$ as constrained by the data-covered periods.



### 2.1.3 The "Explicitly interannual" $p\mathrm{CO_2}$ mapping (E)

In order to stabilize the secular trend in the early decades (as discussed in Sect. A2 below), we now add a fixed (ie., non-adjustable) term (superscript 'fix'),

$$
\begin{aligned}
f_{\mathrm{int}}^{\mathrm{E}}(x,y,t) \;=\; & f_{\mathrm{int,IAV}}^{\mathrm{adj}}(x,y,t) \\
& +\; f_{\mathrm{int,LT}}^{\mathrm{ADJ}}(x,y) + f_{\mathrm{int,Seas}}^{\mathrm{ADJ}}(x,y,s) \\
& +\; f_{\mathrm{int,Decad}}^{\mathrm{fix=OCIM}}(x,y,t)
\end{aligned}
\tag{2}
$$

Consequently, the prior of $f_{\mathrm{int}}^{\mathrm{E}}$ is given by this fixed term. It is obtained from the sea–air flux product by DeVries (2014, updated), which is based on a data-driven model of time-mean ocean circulation (OCIM) applied to the rising atmospheric $\mathrm{CO_2}$ boundary condition. The OCIM fluxes have been decadally smoothed (indicated by subscript 'Decad'), because the OCIM result originally represents sea–air fluxes including SST-related interannual variations, which are created by our parameterizations already (Fig. 1).

### 2.1.4 The "multi-linear Regression" (R)

In the third variant, the ocean-internal DIC flux is represented as

$$
\begin{aligned}
f_{\mathrm{int}}^{\mathrm{R}}(x,y,t) \;=\; & \sum_{i} \gamma_{i}^{\mathrm{adj}}(x,y) \cdot V_{i}(x,y,t) \\
& +\; f_{\mathrm{int,LT}}^{\mathrm{ADJ}}(x,y) + f_{\mathrm{int,Seas}}^{\mathrm{ADJ}}(x,y,s) \\
& +\; f_{\mathrm{int,Decad}}^{\mathrm{fix=OCIM}}(x,y,t)
\end{aligned}
\tag{3}
$$

Compared to Eq. (2), the degrees of freedom representing interannual variations ($f_{\mathrm{int,IAV}}$) are replaced here by a multi-linear function involving three explanatory fields ($V_i$):

 – Sea surface temperature (SST),

 – its temporal change (dSST/dt), and

 – squared wind speed ($u^2$).

There is a two-fold motivation behind this choice of explanatory variables: (1) Variations in carbon-relevant processes (e.g., carbon and nutrient input into the mixed layer, stratification, mixing, entrainment, wind-driven deepening of the mixed layer) are expected to also be related to these variables. (2) Observation-based data sets for SST and $u$ are available over all our calculation period 1951–2021 ($u$ at least from reanalysis). The specific input data sets used in our base case are given in Table 1.

The simultaneous use of SST and dSST/dt is motivated as it is *changes* in SST that are related to DIC fluxes (i.e., *changes* in DIC). Moreover, the sum of SST and dSST/dt mathematically allows a temporal shift between SST and $f_{\mathrm{int}}$ for a dominant Fourier mode (similar to sine and cosine terms).





To prevent confusion, we point out that the multi-linear regression as introduced here is set up in terms of the ocean-internal DIC flux $f_{\text{int}}$ (see Sect. 2.1.1), not in terms of $p\text{CO}_2$ or sea–air flux as done in various other studies in the literature. This also means that important processes (SST-dependence of solubility and carbonate chemistry, wind speed dependence of gas exchange) are not included into the regression Eq. (3), but already taken care of by the parameterizations listed in Sect. 2.1.1 and Fig. 1.

All the explanatory fields $V_i$ are implemented on monthly time scale, smoothed onto our daily time steps. The scaling factors $\gamma_i^{\text{adj}}$ between the internal DIC flux and these explanatory fields $V_i$ are taken as the adjustable degrees of freedom in the cost function minimization (very analogous to the "NEE-T inversion" of Rödenbeck et al., 2018b). These unknown scaling factors are allowed to vary spatially (with correlation length of about $2000\,\text{km}$ in longitude and $1000\,\text{km}$ in latitude, thus more smoothly than the direct adjustments of $f_{\text{int}}$ in the explicitly interannual mapping of Sect. 2.1.3), but are constant in time (1

temporal degree of freedom per explanatory field per pixel). All three regression terms are normalized such that the a-priori uncertainty of their global integral on July 1 (averaged over the July 1 time steps of all years within the analysis period 1957–2020) is the same as that of $f_{\text{int,IAV}}$ in Eq. (2). (July 1 is an arbitrary choice, in line with the normalization with respect to the flux at mid of the final year used in CarboScope so far).

In order to avoid influences of the spin-up transient on the regression coefficients (estimated sensitivities), the regression

terms (first line of Eq. (3)) are only covering the analysis period 1957–2020, while the remaining years before and after are filled by explicitly interannual degrees of freedom just as $f_{\text{int,IAV}}^{\text{adj}}$ in Eq. (2). For clarity, this detail has been omitted from Eq. (3).

In order to explore how robust the results of the multi-linear regression are, we also perform *uncertainty cases* where certain set-up parameters are modified within ranges as plausible as the base case:

**RegrSSTNOAA:** using SST from NOAA_ERSST v5 Huang et al. (2017) as alternative data set for the explanatory variables SST and dSST/d$t$ (but no change to any other SST-dependent items such as solubility);

**RegrU2NCEP:** using wind speeds from NCEP reanalysis (Kalnay et al., 1996) as alternative data set for the explanatory variable $u^2$ (but no change to wind-dependent gas exchange);

**RegrAdddSSTdt2:** Additional regression term $(\text{dSST/d}t)^2$;

**RegrAddU4:** Additional regression term $u^4$;

**RegrAddpaCO2:** Additional regression term $p_a\text{CO}_2$;

**RegrNoDecad:** removing any decadal variability and secular trends from the explanatory fields $V_i$, such that the multi-linear regression term only represents interannual variability on a time scale of a few years;

**RegrShort:** shorter spatial correlation lengths for the sensitivities $\gamma_i^{\text{adj}}$ (supplementary Fig. S5);

**RegrLoose:** a-priori uncertainty of the sensitivities increased by a factor 4 (i.e., the strength of the mathematical regularization is reduced);



**MLDq2:** Dividing mixed-layer depth by 2;

**MLDx2:** Multiplying mixed-layer depth by 2;

**GasexLow:** Weaker gas exchange, by scaling the piston velocity field such that its global mean matches the lower limit of the range $16.5 \pm 3.2\,\mathrm{cm/hr}$ (Naegler, 2009) rather than the central value;

**GasexHigh:** Stronger gas exchange (analoguously, using upper limit);

**GasexU3:** Replacing the $u^2$ dependence of gas exchange by a $u^3$ dependence (while keeping the global mean piston velocity the same).

To help in the discussion of specific aspects, we performed further *test cases* (not necessarily as plausible as the base case):

**RegrOnlySST, RegrOnlydSSTdt, RegrOnlyU2:** The explanatory variables were used individually, i.e., the regression terms
of the remaining two were omitted;

**RegrAddChl_98r19:** Addition of Chl-a as a further explanatory variable (Fig. S7). Chlorophyll concentration has been taken from the GlobColour project (Maritorena et al., 2010), which combined retrievals from the SeaWiFS (NASA), MODIS (NASA), MERIS (ESA), OLCI (ESA), and VIIRS (NOAA/NASA) satellites into a harmonized data set; as the Chl-a data are only available for the years 1998–2019, the regression is restricted to this period (plus spin-up and spin-down
periods);

**RegrHeat_85r09:** replacing dSST/dt with the net sea–air heat flux taken from the OAFlux project (Yu and Weller, 2007), regression period restricted to 1985–2009 according to the availability of the heat flux data set;

**RegrCurl_88r18:** replacing $u^2$ with windstress curl calculated from Cross-Calibrated Multi-Platform (CCMP) v2.0 wind speeds (Atlas et al., 2011), regression period restricted to 1988–2018 according to the availability of CCMP;

**98r19, 85r09, 88r18:** Using the same regression terms as in the base case, but restricting the time period of regression to the same years as used for RegrAddChl_98r19, RegrHeat_85r09, and RegrCurl_88r18, respectively.

### 2.1.5   The "Hybrid" $p\mathrm{CO_2}$ mapping (H)

The final variant aims to combine the temporal extrapolation capability of the multi-linear regression (Sect. 2.1.4) and the flexibility to reproduce observed signals of the explicitly interannual mapping (Sect. 2.1.3). Technically being an explicitly
interannual mapping itself, its representation of the ocean-internal DIC flux,

$$
\begin{aligned}
f^{\mathrm{H}}_{\mathrm{int}}(x,y,t) \;=\; & f^{\mathrm{adj}}_{\mathrm{int,IAV}}(x,y,t) \\
& + \; f^{\mathrm{adj}}_{\mathrm{int,LT}}(x,y) + f^{\mathrm{adj}}_{\mathrm{int,Seas}}(x,y,s) \\
& + \; f^{\mathrm{fix=R}}_{\mathrm{int}}(x,y,t)
\end{aligned}
\tag{4}
$$

is similar to Eq. (2), but with the following changes:





- the result of the multi-linear regression (Sect. 2.1.4) is used as prior for the internal DIC flux ($f_{int}^{fix=R}(x,y,t)$), and

- as the long-term mean and the mean seasonal cycle are now already contained in $f_{int}^{fix=R}(x,y,t)$, they are not adjusted again, i.e., the only degrees of freedom being adjusted in the hybrid run are interannual corrections to the internal DIC flux (however, as the term $f_{int,IAV}^{adj}$ was introduced to denote non-seasonal anomalies only, we still need the terms $f_{int,LT}^{adj}(x,y)$ and $f_{int,Seas}^{adj}(x,y,s)$ to obtain a complete Fourier series, where the lower-case superscript 'adj' indicates that the a-priori uncertainty of these terms is not enhanced, in contrast to Sect. 2.1.2).

Due to this construction, the hybrid result will fall back to the multi-linear regression during periods without data, but it is nevertheless able to fit $pCO_2$ signals on month-to-month, year-to-year, and decadal time scales that have not yet been reproduced via the explanatory variables of the multi-linear regression.

For assessing uncertainties, the hybrid mapping was re-run also with several of the uncertainty cases of the regression as listed in Sect. 2.1.4. Part of the set-up changes (mixed-layer depth, gas exchange) also affect the hybrid calculation itself.

Note that the hybrid run is mathematical equivalent to estimating an additive correction to the multi-linear regression from the $pCO_2$ residuals of the multi-linear regression. That is, the signals being used by the hybrid run are those that could not yet be explained by the multi-linear regression. The hybrid run is thus similar to a hypothetical joint run simultaneously having regression degrees of freedom (like the multi-linear regression) and explicitly interannual degrees of freedom (like the explicitly interannual estimate). We abandoned the concept of such a joint run, however, because it would face two problems: (1) its result would depend on the relative a-priori weighting between the two groups of degrees of freedom, for which there is no clear information; and (2) the explicitly interannual degrees of freedom would necessarily also absorb part of the signals actually proportional to the explanatory variables. Running the multi-linear regression and the hybrid step sequentially, as done here, reduces both problems.

### 2.1.6 The Pre-mapping (P): Determining the linearization of the carbonate chemistry

In contrast to Rödenbeck et al. (2013), we now allow for the secular trend in the Revelle factor. We deem this necessary due to our longer period of interest 1957–2020, during which the mixed-layer carbon content notably increased, leading to shifts in the relation between variations in the ocean-internal DIC flux ($f_{int}$) and the sea–air $CO_2$ flux. As our scheme extrapolates the seasonality (and in the "multi-linear regression" also the interannual variations) from the data-constrained recent decades to the almost unconstrained earlier decades through correlations in $f_{int}$ (see the last paragraph of Sect. 2.1.2), the shifting relation has the potential to alter the amplitude of flux variations in the earlier decades.

As in Rödenbeck et al. (2013), the non-linear dependence of $pCO_2$ on DIC is linearized around reference fields $pCO_{2Ref}$ and $DIC_{Ref}$,

$$pCO_2 = pCO_{2Ref} + \left(\frac{dpCO_2}{dDIC}\right)(DIC - DIC_{Ref}) \tag{5}$$

The linearization is needed to be able to use the fast minimization algorithm in the CarboScope software. Previously in Rödenbeck et al. (2013), the reference fields $pCO_{2Ref}$ and $DIC_{Ref}$ were temporally constant, had been taken from observation-based data sets not guaranteed to be mutually consistent, and the derivative ($dpCO_2/dDIC$) had been calculated from these



via approximation formulas. In order to now include the secular trend in Revelle factor (and simultaneously to remove the mentioned approximations), we employ the *mocsy* package (Orr and Epitalon, 2015), which provides routines to accurately calculate $pCO_2$ and $(dpCO_2/dDIC)$ from a given field of DIC (and from fields of Alkalinity, SST, salinity, silicate, phosphate, and air pressure, which we take from external sources, Table 1). Using an adjusted Newton algorithm calling *mocsy* iteratively,

we obtain an algorithm to calculate (reference) DIC and $(dpCO_2/dDIC)$ from a given (reference) $pCO_2$ value at each location and time (box L in Fig. 2). The $pCO_{2\text{Ref}}$ field is obtained as the posterior $pCO_2$ field of a "pre-mapping" run (P, the leftmost one in Fig. 2). The $pCO_{2\text{Ref}}$ and $(dpCO_2/dDIC)$ fields used in this pre-mapping run, in turn, are calculated from a preliminary reference identical to atmospheric $pCO_2$. This yields a reasonable starting point, because the atmospheric $pCO_2$ field does already contain the secular $CO_2$ rise, which is the most important feature in this context.

Potentially, we might expect to need a loop with further pre-mappings, each getting its $pCO_{2\text{Ref}}$ field from the posterior $pCO_2$ field of the respective previous one. However, we confirmed by explicit testing that the fields are not appreciably altered any more after the first pre-mapping; thus a single pre-mapping is sufficient.

All other mapping runs of this study use the same spatio-temporal linearization fields $pCO_{2\text{Ref}}$, $DIC_{\text{Ref}}$, and $(dpCO_2/dDIC)$ as calculated by the pre-mapping.

### 2.1.7  Technical details common to all variants

In contrast to Rödenbeck et al. (2013), the calculation period of all runs starts in 1951. This allows the initial transient of the mixed-layer DIC budget equation to decay by 1957 (start year chosen in light of the potential use of the results as prior in atmospheric inversions). As in Rödenbeck et al. (2013), the calculation period includes one more year ("spin-down", here 2021) after the valid period constrained by the data (until end of 2020), in order to avoid numerical edge effects.

In order to cover all the calculation period since 1951, we now use SST from Hadley EN.4.2.1 (Good et al., 2013) and sea ice concentration from HadISST 2.2.0.0. (Titchner and Rayner, 2014, accessed on 2020-06-05 from https://www.metoffice. gov.uk/hadobs/hadisst2/data/HadISST.2.2.0.0_sea_ice_concentration.nc.gz).

Compared to Rödenbeck et al. (2013), the spatial resolution of all the mapping calculations has been increased to 2.5° longitude × 2° latitude (previously on the grid of the TM3 atmospheric transport model, 5° × 4°). Moreover, the adjustments
are now done over the entire ocean (i.e., we do not fix part of the temporally ice-covered regions any more).

### 2.2  Gauging the predictive skill of the multi-linear regression

In order to test whether the multi-linear regression against explanatory variables (Sect. 2.1.4) is actually meaningful, we determine its predictive skill. For this, the multi-linear regression is re-run 6 times, each time omitting the $pCO_2$ data from one of the 5-year periods 1985–1989, 1990–1994, 1995–1999, 2000–2004, 2005–2009, or 2010–2014. That is, each of the 6 test
runs possesses an artificial data gap of 5 years, a duration chosen to be longer than typical features of year-to-year variability like El Niño. We can then compare the predictions during the data gaps with the results of the completely constrained run.





## 3 Results

The main results of this study are of two different types:

- From the *multi-linear regression*, we obtain spatial maps of the sensitivities $\gamma_i$ (Eq. (3)) relating the variations in the surface-ocean carbon system to variations in SST, dSST/d$t$, and $u^2$.

- The *hybrid mapping* yields a spatio-temporal estimate of the sea–air $CO_2$ flux over 1957–2020, in particular its evolution from year to year.

For easier exposition, we start with a time-series view of the sea–air $CO_2$ flux, and only then present the contributions of the various environmental conditions to this variability as determined by the regression.

### 3.1 Sea–air $CO_2$ flux variations estimated by the hybrid mapping

Fig. 3 shows estimated interannual (i.e., slower-than-seasonal) variations of the sea–air $CO_2$ flux, subdividing the ocean into basins and latitude bands. We first focus on the hybrid mapping (blue) as our most comprehensive mapping variant (Sect. 2.1.5). The most prominant feature of interannual variability is the secular trend towards more $CO_2$ uptake in all ocean regions. Considering variations around this secular trend, the tropical Pacific is the region providing the largest contribution to total ocean variability (compare Quéré et al., 2000), both on decadal time scale and on year-to-year time scale. The year-to-year

variations are strongly tied to El Niño as indicated by the background stripes (Feely et al., 1999). When considering trends within individual decades, the decadal increase in the $CO_2$ sink slowed down in the 1990s and early 2000s, and accelerated again afterwards (Landschützer et al., 2016; DeVries et al., 2019), even though these decadal trends seem to be the consequence of pronounced anomalies on the faster year-to-year time scale rather than representing actual slowlier decadal variations.

### 3.2 How do the year-to-year sea–air $CO_2$ flux variations estimated by regression and hybrid mapping compare with

20        each other?

In addition to the variations of the sea–air $CO_2$ flux estimated by the hybrid mapping (blue), Fig. 3 also shows those estimated by the multi-linear regression (Sect. 2.1.4, orange) and the explicitly interannual $pCO_2$ mapping (Sect. 2.1.3, green). From the late 1980s onwards, when progressively more $pCO_2$ data are available to constrain interannual variations explicitly, the hybrid mapping (blue) shows some corrections over the multi-linear regression (orange). For the large El Niño-related variability in

the tropical Pacific, these corrections are generally small compared to the estimated variations themselves. This indicates that the multi-linear regression already captures a notable fraction of the year-to-year flux variations in this region, even though it underestimates the size of most of these anomalies (the interannual standard deviation 1985–2019 from the multi-linear regression is only about 82% of that from the hybrid mapping in the tropical Pacific). Fig. 4 (dots) confirms that the hybrid mapping fits the $pCO_2$ data closely (the blue dots are located right under the black dots), while the multi-linear regression

(orange dots) also follows the variability in the data (black dots) but does not match them as closely as the hybrid mapping.





In the intermediate and high latitudes (top and bottom panels of Fig. 3), in contrast, the multi-linear regression (orange) does not pick up most of the year-to-year anomalies. This may indicate that the set of explanatory variables used in the regression misses essential modes of variability there. On the other hand, some of the variations estimated with explicitly interannual degrees of freedom (green and blue) may also be spurious effects from the temporally very uneven data coverage.

Although the hybrid mapping (blue) has the same interannual degrees of freedom (i.e., the same flexibility) as the explicitly interannual mapping, it does not always bring the fluxes back to the explicitly interannual result (green), especially in the region south of the tropical Pacific (Fig. 3). Since the two estimates are actually very close to each other where data exist (as illustrated in Fig. 4: the green dots are essentially invisible under the co-located blue and black dots, despite the differences between the green and blue lines), the differences in areal averages as in Fig. 3 reflect differences in data-void areas/periods

being filled by the mappings. However, while the explicitly interannual mapping falls back to the unconstrained prior, the hybrid mapping falls back to the multi-linear regression which is at least indirectly constrained via the statistical relationships between the ocean-internal DIC flux and the chosen explanatory variables (Sect. 3.3 below). This may also prevent some undue spatial extrapolation from the tropical Pacific into unconstrained areas by the explicitly interannual scheme. Thus, we expect the hybrid mapping (blue) to be more realistic than either of the multi-linear regression and the explicitly interannual mapping.

### 3.3   How much predictive skill does the multi-linear regression have?

The results of the multi-linear regression (and thus the hybrid mapping during poorly constrained periods) are only meaningful if the regression actually possesses some predictive skill to bridge unconstrained periods. In order to test this, we performed runs with artificial data gaps of 5 years length (Sect. 2.2). Fig. 5 illustrates this using runs discarding all $pCO_2$ data during 1995–1999. As the explicitly interannual mapping draws all information about year-to-year variations from the data and therefore

does not have any predictive skill, it essentially defaults to the prior (having upside-down El Niño response as it misses any variations related to the ocean-internal sources and sinks) during the data gap (Fig. 5 left), except for a shift in long-term mean (see Sect. 2.1.2, last paragraph, for explanation). In contrast, the multi-linear regression (middle) almost completely reconstructs the 1995–1999 flux variations, based on the relationships between the ocean-internal DIC flux and the driving variables learned on the basis of the remaining data outside 1995–1999.

As demonstrated by Fig. S1 in the supplement, this predictive skill holds generally for all parts of the ocean and other 5-year data gaps. This means that no particular $pCO_2$ data point is causing features in the variability and the estimated sensitivities (Sect. 3.5 below) by its own.

In view of applying the multi-linear regression as a prior of the hybrid mapping, the predictive skill is only meaningful to the extent that the multi-linear regression is actually able to explain all signals in the data. For example, since the regression

underestimates the year-to-year anomalies in the tropical Pacific compared to the explicitly interannual estimate as discussed above, it will fill data gaps with too small an amplitude (Fig. 5, right). Even though this is still a clear qualitative improvement compared to the explicitly interannual mapping (left), it indicates that the variability extrapolated into the earlier decades without data will likely be underestimated, too.





### 3.4 How do the three explanatory variables contribute to the interannual variations estimated by the multi-linear regression?

When disregarding the secular increase in the ocean carbon sink, the largest year-to-year variations of the regionally integrated sea–air carbon flux are found in the tropical Pacific (Fig. 3). Correspondingly, the year-to-year variations in the ocean-internal

carbon flux ($f_{\text{int}}$) from the three terms in the multi-linear regression (Eq. (3)) are largest in the tropics as well (Fig. 6, left). Of the three explanatory variables, the contribution of year-to-year variability in temporal SST changes (dSST/d$t$, black) is the largest. Concurrent with the warming (dSST/dt>0) at the onset of each El Niño event (grey background stripes), we find a negative carbon flux anomaly (reduction of the carbon source in this region) because smaller amounts of cold, carbon-rich water are upwelling. At the end of each El Niño event, we find an analoguous coupling of the cooling (dSST/dt<0) and an additional

carbon source to the mixed layer. The contribution of year-to-year variability in SST itself (red) is second largest in the tropics, causing anomalous carbon sinks during El Niño events and anomalous carbon sources during La Niña conditions afterwards. This could be interpreted as a small correction to the dSST/dt contribution: the sum of the dSST/dt and SST contributions (not shown) is similar to the dSST/dt contribution alone, but slightly shifted in time by a few months. The smallest contribution to the year-to-year variability in the tropics is estimated for squared wind speed ($u^2$, light blue), with a temporal pattern relatively

similar to that of the SST contribution. Due to the co-variation between SST and $u^2$ on year-to-year time scale, these two explanatory variables could be partly confounded by the regression, though their respective spatial patterns are different (see below).

    In the high-latitude bands, the wind speed contribution is estimated to be larger than in the tropics, now on the same order of magnitude as the SST and dSST/dt contributions or even larger. As a notable feature in the Southern Ocean (bottom left panel

of Fig. 6), the secular increase in wind speed leads to a secular increase in the carbon source into the mixed layer. Across our set of uncertainty cases listed in Sect. 2.1.4, the linear trend of the wind speed contribution over the 1960–2019 period in the ocean south of $45^\circ\,S$ is estimated in the range 0.002 to $0.005\,(\text{PgC}\,\text{yr}^{-1})\,\text{yr}^{-1}$ (see supplementary Fig. S3, bottom, light blue bars). As a secular trend in $f_{\text{int}}$ (bottom left in Fig. 6) causes a secular trend in sea–air flux of the same size (bottom right), it represents a reduction by 17 to $42\%$ of the increase in the Southern Ocean sink strength (relative to the trend of $-0.012\,(\text{PgC}\,\text{yr}^{-1})\,\text{yr}^{-1}$

estimated by OCIM). A slowing-down of the Southern Ocean sink increase (compared to the increase expected from rising atmospheric $CO_2$) has also been found in model simulations and attributed to an increase in upwelling of old carbon by the accelerating winds (Le Quéré et al., 2007; Hauck et al., 2013, and many others). We need to note, however, that our multi-linear regression estimates the wind-speed related trend only indirectly: As the sensitivites $\gamma_{u^2}$ are presumably largely constrained by year-to-year variations (because they do not change much if the linear trend of the explanatory variables is removed, see

sensitivity case RegrNoDecad, Sect. 4.3), the slope of the secular trend can only be correct to the extent that the sensitivity $\gamma_{u^2}$ is identical for year-to-year and secular variations.

    The year-to-year anomalies from the $f_{\text{int}}$ contributions (Fig. 6, left) carry through to the sea–air $CO_2$ flux (right) in a delayed and dampened fashion, due to the buffer effect of carbonate chemistry in combination with the limited gas exchange. We also





note again that the sea–air $CO_2$ flux contains additional year-to-year variability from solubility and gas exchange anomalies as represented by the involved parameterizations (also see Fig. 1).

### 3.5 Patterns of the sensitivity of ocean-internal DIC sources and sinks to interannual variations of SST, dSST/dt, and $u^2$ estimated by the multi-linear regression

The estimated sensitivities of the ocean-internal DIC flux ($f_{int}$) against interannual variations in the chosen explanatory variables of the multi-linear regression (sea surface temperature SST, temporal changes in sea surface temperature dSST/dt, and squared wind speed $u^2$) are shown in Fig. 7. Here we consider the most prominent features in these sensitivity patterns and mention oceanic processes that are compatible with these and may thus control surface-ocean biogeochemistry. Even though regression analysis cannot proof causation, we will argue later (Sect. 4.1) why such a tentative attribution may be meaningful

here. Also see Sects. 4.3–4.5 for further discussion on uncertainties.

#### 3.5.1 Sensitivity of $f_{int}$ to dSST/dt

We start with dSST/dt (Fig. 7, top) as the explanatory variable contributing the largest year-to-year variability (Sect. 3.4). Events of decreasing SST are estimated to be associated with more positive ocean-internal DIC fluxes in the tropical Pacific (within a tilted band located around the equator in the western tropical Pacific and around about $15°S$ in the eastern tropical

Pacific) and in most parts of the higher latitudes in both hemispheres (blue and cyan areas in Fig. 7 top). Such a correlation would arise from variations in the upwelling of waters that are both colder and more carbon-rich than the mixed layer.

    In the rest of the ocean, the absolute value of the sensitivity $\gamma_{dSST/dt}$ is small (light blue or light red). We assume that these sensitivities mainly reflect insignificant correlations, especially due to the higher uncertainty in regions of sparse data coverage or in regions where dSST/dt is mainly driven by atmospheric heating or cooling. In particular, positive sensitivities are not

compatible with any known oceanic mechanism.

#### 3.5.2 Sensitivity of $f_{int}$ to SST

The estimated sensitivity $\gamma_{SST}$ between the interannual varitions of the ocean-internal DIC flux and SST itself is rather patchy, with both positive and negative areas (Fig. 7, middle). This may reflect that various biological processes contribute to $f_{int}$, depending on temperature in different ways and thus potentially cancelling each other. For example, carbon fixation (Net

Primary Productivity, NPP) will envigorate with increasing temperature (until a threshold is reached); as NPP represents a sink (i.e., a negative contribution to $f_{int}$), it would thus cause negative $\gamma_{SST}$ sensitivities. Carbon export (or export ratio at least) is generally anticorrelated with temperature (Laws et al., 2000), thus causing positive $\gamma_{SST}$ sensitivities, though also the opposite behaviour seems possible.

    Positive interannual sensitivity to SST would also be compatible with a nutrient effect. Upwelling both decreases SST and

increases the availability of nutrients. Thus, negative anomalies in SST tend to be associated with higher biological production, thus enhanced removal of carbon (negative anomalies in $f_{int}$). However, upwelling also brings up carbon, which is usually



assumed to dominate the carbon signal. For example, Hauck et al. (2013) showed that –in the model– in the Southern Ocean south of 55° S, there would be more biological export per increase in the Southern Annular Mode (SAM) which goes along with more upwelling. Yet, this is still controversially discussed.

As the statistical inference by our regression can only respond to the sum of all contributing processes, we therefore cannot
draw specific conclusions from the estimated $\gamma_{SST}$ pattern. In addition, the regression may adjust $\gamma_{SST}$ to effectively shift the dSST/dt contribution in time (Sect. 3.4).

### 3.5.3  Sensitivity of $f_{int}$ to $u^2$

Higher wind speeds are estimated to be associated with more positive ocean-internal DIC fluxes (stronger sources into or weaker sinks out of the mixed layer) along the equator in the Pacific, in the eastern upwelling zones of the North Pacific, South
Pacific, and South Atlantic, as well as in circumpolar bands in the high latitudes of both hemispheres (red and yellow areas in Fig. 7 bottom). Such a positive sensitivity is compatible with wind-driven deepening of the mixed layer, Ekman pumping, or speeding up of the wind-driven upwelling, such that more carbon-rich waters are mixed in from below during stronger winds.

In contrast, higher wind speeds tend to be associated with more negative ocean-internal DIC fluxes (i.e., weaker sources or stronger sinks) at the western extratropical fringes of all ocean basins (blue areas). In these regions of mode water formation,
higher wind speeds lead to more subduction of anthropogenic $CO_2$ away from the surface into the ocean interior.

### 3.5.4  Additional variability of the sea–air flux

We note again that the sensitivities discussed here are those of the ocean-internal DIC sources and sinks $f_{int}$ (Fig. 1, bottom panel). The sea–air $CO_2$ fluxes contain additional variability also driven by interannual variations in SST (e.g., via the changes in $CO_2$ solubility) or in wind speed (via the piston velocity of gas exchange). As this additional variability is already generated
by the parameterizations contained in our algorithm (Sect. 2.1.1), these processes are, within uncertainties, not reflected in the sensitivities against SST or $u^2$ again.

## 4  Discussion: Robustness of the multi-linear regression

### 4.1  How meaningful is the multi-linear regression in terms of biogeochemical processes?

Statistical inferences by multi-linear regression are at the risk of overfitting, i.e. adjustment of coefficients to follow minor
signals in the data, or even noise. If that was the case, the estimated sensitivities would not reflect underlying biogeochemical processes. Various findings indicate however that the results of the presented multi-linear regression do reflect actual signals:

–  The patterns of the sensitivities (Fig. 7), at least those with respect to dSST/dt and $u^2$, are quite systematic spatially, and in many respects interpretable (Sect. 3.5). This is especially true in the tropical Pacific, but also throughout the entire ocean.





- – Test regression runs only using a single one of the explanatory variables yield sensitivities very similar to the base case using all explanatory variables (supplementary Fig. S2). This indicates that the regression terms are essentially mutually independent, such that each explanatory variable picks up a more or less unique portion of the signals contained in the data.

- – The regression possesses predictive skill (Sect. 3.3 above), which also means that it does not depend on any particular portion of the $pCO_2$ data or the explanatory fields alone.

- – The estimates are relatively robust against alternative data sets for the explanatory variables (cases RegrSSTNOAA and RegrU2NCEP in supplementary Fig. S4). This corroborates that the regression is likely not dominated by any particular feature in these fields.

- – The regression hardly responds to a 4-fold change in the regularization strength (case RegrLoose in supplementary Fig. S4). In the overfitting regime, one would expect a substantial dampening effect when the regularization is stronger.

- – The regression results are quite robust against further changes in the set-up (supplementary Figs. S4, S5, and S6).

The relatively small set of explanatory variables used here (Sect. 2.1.4) is certainly helpful to avoid overfitting (compare, e.g., Thacker, 2012). For example, in test runs with seasonally resolved (rather than temporally constant) sensitivity coefficients,
the data could be fitted more closely, but the predictive skill deteriorated (not shown).

### 4.2 Which fraction of the year-to-year variability can be captured by the multi-linear regression?

As seen in Fig. 3 and quantified explicitly in Fig. 8 (top), the amplitude of year-to-year variability in the global sea–air $CO_2$ flux estimated by our multi-linear regression (orange) is lower than that estimated by the hybrid mapping possessing the degrees of freedom to follow any interannual signals (blue). This indicates that the $pCO_2$ data also contain signals of year-to-
20  year variability that cannot be represented in terms of the variations contained in the set of explanatory variables used in the regression. Possibly, however, the hybrid mapping may also exaggerate the amplitude of signals by spreading them over too large an area in data-poor parts of the ocean.

The situation is different in the tropical Pacific (Fig. 8, bottom). Here, the multi-linear regression (orange) already captures a large part of the variability found in the hybrid mapping (blue). This indicates that our explanatory variables are reasonably
suited to represent the ENSO-related variability dominating in this region.

To elucidate the ability of the multi-linear regression to capture year-to-year anomalies, we compare it with other $pCO_2$ mappings based on linear or non-linear regressions of $pCO_2$ (itself) against various sets of explanatory variables (Landschützer et al., 2013; Iida et al., 2020; Denvil-Sommer et al., 2019; Gregor et al., 2019). Globally (Fig. 8, top), the variability obtained by the other $pCO_2$ mappings (salmon) is larger than that from our multi-linear regression (orange). Closer inspection (not
shown) reveals that these larger amplitudes mostly reflect variability on multi-year (decade-to-decade) time scales occurring coherently in both northern and southern extratropics, while the multi-linear regression does not inolve such globally correlated contributions. Accordingly, when splitting up the global flux into regional contributions, the amplitudes from the other $pCO_2$





mappings and our multi-linear regression are quite comparable. For example, in the tropical Pacific (Fig. 8, bottom) our regression yields year-to-year variability larger than any of the other $pCO_2$ mappings considered. Based on reconstructions of model-based pseudo data, Gloege et al. (2021) found for one of the other methods included in Fig. 8 that the amplitude of Southern Ocean decadal variability was overestimated by $15$ to $58\%$.

Could alternative or additional explanatory variables help to capture a larger fraction of variability by the multi-linear regression?

- As the explanatory variables of the base case are all physical variables, we tested using chlorophyll-a concentration as a biological variable (run RegrAddChl_98r19, supplementary Fig. S7). A practical problem with chlorophyll-a is that data sets are only available for the most recent years (from 1998); therefore it is not used in our base case. The test
suggests, however, that chlorophyll is not actually adding much information about the year-to-year variations in the sea–air $CO_2$ flux beyond what is already provided by the explanatory variables of the base case (SST, dSST/dt, and $u^2$). A reason may be that chlorophyll variability is already covered in the other variables, as nutrients are also a function of upwelling and stratification etc. It is also important to keep in mind that chlorophyll concentration is not directly observed but indirectly inferred from optical properties of the seawater, such that the chlorophyll data may contain
substantial variability unrelated to the carbonate system.

- Conceivably, more general non-linear relationships between $pCO_2$ and the explanatory variables may allow to capture signals not represented in linear relationships as used in our base case. Uncertainty cases involving additional regression terms proportional to $(dSST/dt)^2$ (run RegrAdddSSTdt2) and $(u^2)^2$ (run RegrAddU4), respectively, only marginally increase year-to-year variability (within the narrow band in Fig. S4). Also from the set of other $pCO_2$ mappings (salmon
in Fig. 8), there is no indication that the non-linear regressions (CMEMS-FFNN, CSIR-ML6, and MPI-SOMFFN) would generally capture more variability than the linear ones (JMA-MLR and ours). We conclude that non-linearities in the $pCO_2$ relationships are not essential for explaining year-to-year anomalies in the $pCO_2$ field on regional scale.

- Using heat flux as explanatory variable instead of dSST/dt (RegrHeat_85r09) deteriorates the ability of the multi-linear regression to reproduce ENSO-related variability (supplementary Fig. S8).

- Replacing $u^2$ by the windstress curl (RegrCurl_88r18) does not change the flux IAV much (supplementary Fig. S10). A further alternative explanatory variables may be "Ekman pumping", which however diverges at the equator, and was not tested.

A common methodological feature of all present-day regression-based $pCO_2$ mappings including ours is that the carbon variables are only related to the concurrent values of the explanatory variables, disregarding any dependence on past values
of the explanatory variables possible due to memory effects. This might be a serious limitation, but allowing for memory effects is not straightforward. For example, regression terms with lagged explanatory variables would only allow discrete lag times, and using an extensive spectrum of lag times would possibly exceed the number of well-determined degrees of freedom. Theoretically, fitting comprehensive process models (GOBMs) to the $pCO_2$ data would include emerging memory





effects, but this faces various conceptual and computational challanges (see a recent application of a low-dimensional Green's function approach by Carroll et al., 2020). (Note that the amplitudes simulated by the hindcast GOBMs included in Fig. 8 are roughly similar to those from our multi-linear regression and smaller than those from the hybrid scheme.) If memory effects are important, they are evidently not yet adequately captured by those elements in our algorithm that do represent history effects

(the budget equation accumulating past $f_{\text{int}}$ contributions, the seasonal "history flux" (Rödenbeck et al., 2013), and the use of both SST and dSST/d$t$ as explanatory variables).

### 4.3 To which extend depend the sensitivities $\gamma_i$ on time-scale?

In our formulation of the regression (Eq. (3)), the sensitivities $\gamma_i$ are applied to the fields $V_i$ of the explanatory variables including all their variations on year-to-year, decadal, and secular time scales. Conceivably, however, the relationships between

$f_{\text{int}}$ and the explanatory variables may differ for year-to-year, decadal, or secular variations. In ocean areas where the data period is long enough to possibly constrain decadal time scales directly, the estimates may therefore reflect some mixture of time scales, which would be hard to interpret.

We assessed this by the uncertainty case RegrNoDecad where any decadal variability (including any secular trend) has been removed from the 3 explanatory variables. As this case can only pick up year-to-year signals to constrain the sensitivities,

any changes compared to the base case may indicate such potential time scale conflicts. In most regions, this is not evident (supplementary Fig. S6). Exceptions are the southern Pacific and the tropical Indian (for the wind-speed sensitivity $\gamma_{u^2}$) and the western tropical Pacific (for the SST sensitivity $\gamma_{\text{SST}}$). As the explanatory variable dSST/d$t$, which dominates the large tropical variability, does not have much secular trend, it is not prone to time-scale dependence anyway.

An alternative way to assess the impact of secular trends in the explanatory variables is the uncertainty case RegrAddpaCO2

having an additional regression term proportional to atmospheric $CO_2$ ($p_a CO_2$). As $p_a CO_2$ is rising steadily over the calculation period, this run is able to adjust the secular trend independently of the trends in SST, dSST/d$t$, or $u^2$, thus breaking any potential time scale conflicts. Indeed, the sensitivities estimated by RegrAddpaCO2 (not shown) are similar to those from the base case as well, and any differences from the base case are similar to those of RegrNoDecad.

We note that in ocean areas with data periods of a few years only, a possible time-scale dependence will not affect the

sensitivities themselves, but it may still affect secular trends in the fluxes if sensitivities estimated for year-to-year variations are applied to secular trends in the explanatory variable. We do not have a means to detect whether this is the case.

### 4.4 Spurious effects from uncertainties in the parameterizations

Errors in the sea–air $CO_2$ flux resulting from deficiencies in our chosen parameterizations of solubility and gas exchange lead to compensating spurious contributions to $f_{\text{int}}$, because it is the sum of both fluxes which changes the mixed-layer carbon

content in our budget equation (see Fig. 1 or  Rödenbeck et al., 2013). This will then also lead to spurious contributions to the estimated sensitivities $\gamma_i$. For example, spurious $u^2$ sensitivity may arise if the wind speed dependence of our gas exchange parameterization is not strong enough such that it is re-inforced by additional changes in the ocean-internal carbon flux (or vice versa).





Luckily, the interannual variability of the sea–air $CO_2$ flux is much smaller than that of $f_{int}$ due to the buffer effect (see Fig. 1). Therefore, in relative terms, the error in the sea–air $CO_2$ flux translates into a much smaller error in $f_{int}$ and in the sensitivities $\gamma_i$.

### 4.5 Spurious effects from missing interannual alkalinity variations

5  The estimated ocean-interanl DIC flux $f_{int}$ –and thus the estimated sensitivities $\gamma_i$ in the regression– contain some spurious contributions to compensate any errors in our representation of carbonate chemistry, because the $pCO_2$ data constrain the $pCO_2$ field rather than the DIC field (Fig. 1). Even though we represent the carbonate chemistry –up to the linearization– by exact equations (Sect. 2.1.6), some error arises because we only use a seasonal alkalinity climatology, while alkalinity also varies interannually due to (1) changing degrees of dilution due to freshwater fluxes (evaporation, precipitation, ice formation, 10  and ice melt), as well as (2) mixing-in of alkalinity-rich deep waters.

(1) Freshwater fluxes do not only dilute alkalinity but also DIC, in equal proportions. On the other hand, the sensitivities of $pCO_2$ against changes in alkalinity and DIC are almost equal in absolute value but of opposite sign (Sarmiento and Gruber, 2006). Therefore, the total effect of freshwater fluxes on $pCO_2$ is small compared to that on alkalinity and DIC, respectively. Therefore, as we neglect both the freshwater contributions to $f_{int}$ and the freshwater-related alkalinity variations, the combined 15  error in $pCO_2$ should be small.

(2) Alkalinity variations related to mixing from below are linked to DIC variations as well, because deep waters are rich in both DIC and alkalinity, compared to the mixed layer. In contrast to the freshwater effects, however, the regression terms $\gamma_i V_i$ in Eq. (3) do contain mixing contributions to $f_{int}$, such that the absence of the corresponding alkalinity variations does affect our $pCO_2$ field being matched to the data. On the seasonal time scale (where there is no problem anyway as we are using a monthly 20  alkalinity climatology), alkalinity variations in the tropical and subtropical oceans are dominated by freshwater effects; only at higher latitudes, alkalinity variations are increasingly affected by mixing (Lee et al., 2006). For the interannual time scales relevant here, the relative role of mixing is unclear. A better understanding –and hopefully solution– of this problem remains for further work.

We note that the spurious compensatory contributions to $f_{int}$ do not affect the $pCO_2$ field being constrained by the observations. 25  Thus, they essentially do not affect the estimated sea–air $CO_2$ fluxes either.

### 4.6 How well constrained is the secular flux trend?

As discussed in more detail in the appendix (Sect. A2), the secular trend is mostly determined through the prior derived from the OCIM estimate based on ocean-interior data (DeVries, 2014, updated). Due to the lack of $pCO_2$ data in the early decades, the mapping does not add credible information about the secular trend. Due to some slight inconsistency in the use of the prior, 30  the secular trend is even slightly overestimated (Sect. A2); this remains to be addressed in future versions of our flux product.



## 5  Conclusions

In this study, we considered the interannual variability of the sea–air $CO_2$ flux over the 1957–2020 period, constrained by the $pCO_2$ measurements from the SOCATv2021 data base (Bakker et al., 2016). Extending the $pCO_2$ mapping scheme of Rödenbeck et al. (2013, 2014), we employed (1) a multi-linear regression against interannual anomalies of sea surface

temperature (SST), the temporal changes of SST (dSST/d$t$), and squared wind speed ($u^2$), and (2) a subsequent explicitly interannual additive correction, yielding a "hybrid" estimate of spatio-temporal variations in the contemporary sea–air $CO_2$ flux.

– According to our multi-linear regression, interannual variability in the tropical Pacific is dominated by a positive correlation of ocean-internal DIC fluxes to dSST/d$t$, as arising from variations in the upwelling of colder and more carbon-rich

waters into the mixed layer.

– In the eastern upwelling zones as well as in circumpolar bands in the high latitudes of both hemispheres, we find a positive sensitivity to wind speed, compatible with the entrainment of carbon-rich water during wind-driven deepening of the mixed layer. To the extent that this sensitivity inferred from year-to-year variations also applies to secular trends, the wind trend in the Southern Ocean (south of 45 ° S) implies a wind-related reduction of the flux trend by about 17 to

42% (less strong increase in sink).

– As a $pCO_2$ mapping method, the hybrid mapping combines (a) the ability of regression to bridge data gaps and extrapolate into the early decades without much $pCO_2$ data constraint and (b) the ability of an autoregressive interpolation to follow signals even if not represented in the chosen set of explanatory variables. This way, at least the large contributions of the tropical Pacific to the global year-to-year variability of the oceanic $CO_2$ exchange can be extrapolated over all the

1957–2020 period, even though the extrapolated variability prior to about 1985 is probably underestimated.

## Appendix A:  The global ocean carbon sink estimated by the hybrid mapping

Here we discuss the global total of the sea–air $CO_2$ flux as estimated by the hybrid mapping, and compare it to various literature estimates. In order to allow a quantitative comparison, we focus on specific features, namely the mean flux (Sect. A1) and the secular flux trend (Sect. A2).

**A1   The mean sink (1994–2007)**

Fig. 9 shows the contemporary global sea–air $CO_2$ flux estimated by the hybrid mapping (blue solid bar) averaged over the 1994–2007 period. According to the set of uncertainty cases shown (blue hashed bars), the uncertainty in the mean flux from the hybrid mapping is dominated by the uncertainty of gas exchange (cases GasexLow, GasexHigh, and GasexU3; diagonally hashed bars), while all other uncertainty cases do not affect the mean sink estimate very much.





The spread between the flux estimates from other $pCO_2$ mapping methods (group of salmon bars) together with the base case of our hybrid mapping (solid blue bar) only indicates uncertainties due to the mapping algorithms, as all the estimates use the same global scaling of the piston velocity from Naegler (2009). Notably, this spread does not exceed the differences due to scaling sea–air gas exchange within the uncertainty range of Naegler (2009) (cases GasexLow, GasexHigh).

The comparison between the results of the hybrid mapping and further literature values is hampered as $pCO_2$ mappings are estimating the total contemporary net $CO_2$ flux ($F_{net}$) through the sea–air interface, while other methods may only include certain components of it. Adopting the notation by Hauck et al. (2020), Table 3 gives the six components of $F_{net}$ and their respective inclusion in the literature estimates considered here (note that the terms "anthropogenic" or "contemporary" are also defined differently in part of the literature).

From the increase in the anthropogenic carbon inventory in the ocean between the extensive ocean surveys in 1994 and 2007, Gruber et al. (2019) estimate an anthropogenic $CO_2$ uptake of $F_{ant,ss} + F_{ant,ns} = -2.6 \pm 0.3\,\mathrm{PgC\,yr^{-1}}$ over the interjacent period, shown in Fig. 9 as long-dashed line. This estimate conceptually differs from the hybrid mapping by the river-induced flux $F_{riv,ss} + F_{riv,ns}$ and the non-steady state modifications $F_{nat,ns}$ to the natural sea–air fluxes, while $F_{nat,ss}$ is zero at the global scale (Table 3). The river-induced flux is very uncertain, with literature estimates ranging between $0.45 \pm 0.18\,\mathrm{PgC\,yr^{-1}}$
(Jacobson et al., 2007) and $0.78 \pm 0.41\,\mathrm{PgC\,yr^{-1}}$ (Resplandy et al., 2018), though the real uncertainty may be even larger. If the Gruber et al. (2019) estimate is shifted by a mid-range river-induced value of $0.62\,\mathrm{PgC\,yr^{-1}}$ (resulting in the dotted line), the base-case value from the hybrid estimate is matched more closely. Nevertheless, given the uncertainty ranges of gas exchange, river-induced outgassing, and the Gruber et al. (2019) estimate, we cannot draw any conclusions from the remaining difference.

The $CO_2$ flux difference between the hybrid estimate and the dotted line in Fig. 9 may also contain a contribution from systematic differences between $pCO_2$ in the bulk ocean water (as typically measured at a few meters depth) and $pCO_2$ at the diffusive surface layer (as relevant for gas exchange), arising due to systematic differences in water temperature and salinity (Woolf et al., 2016). Further, the cooler ocean skin temperature translates the atmospheric $pCO_2$ to a different concentration than that implicitly calculated based on bulk temperature (Robertson and Watson, 1992). Watson et al. (2020) estimated that the
sum of these two effects would shift $pCO_2$-based estimates of the mean global $CO_2$ flux by $-0.8$ to $-0.9\,\mathrm{PgC\,yr^{-1}}$ (stronger sink). So far, however, it is unclear how well the water temperature at the relevant vertical positions can actually be determined (an important source of uncertainty not included in Watson et al. (2020)'s range) and how it varies in space and time. In any case, we note that our study mainly considers the variability of the flux, for which the effect of a time-constant correction as in Watson et al. (2020) would cancel out.

Fig. 9 further shows the global fluxes simulated by a set of Global Ocean Biogeochemical Models (GOBMs) collated in the annual Global Carbon Budget (Friedlingstein et al., 2020, mint green). Like OCIM or Gruber et al. (2019), the GOBMs results do not include the river-induced flux component, but they do conceptually include the non steady-state modification $F_{nat,ns}$ of carbon uptake and natural carbon cycling (Table 3). The range of results covered by the GOBMs slightly exceeds the range of the hybrid estimates due to the gas exchange uncertainty. The median of the GOBM ensemble and the $pCO_2$ mapping
ensemble differ by less than the mid-range river-induced value of $0.62\,\mathrm{PgC\,yr^{-1}}$.





## A2 The secular sink trend (1960–2019)

Regarding the 1960–2019 secular sink trend, our estimate from the hybrid mapping (1) is not able to add much independent information, and (2) even slightly overestimates this trend relative to OCIM used in the prior:

(1) According to Fig. 10 (top), the 1960–2019 trend from the base case (blue solid bar) is quite similar to that of the base-case

prior (grey open bar). Among the uncertainty cases (blue hashed bars), largest deviations are seen when mixed-layer depth is changed (MLDq2 and MLDx2); these deviations are in fact mostly inherited from their respective priors as well (not shown).

(2) Fig. 10 (top) further reveals that the prior (grey open bar) has a slightly steeper trend than the OCIM estimate (magenta), even though the prior has been derived from OCIM (Sect. 2.1.3). This discrepancy arises because we are using OCIM's sea–air fluxes as a prior of the ocean-internal flux $f_{\mathrm{int}}$, even though these two quantities differ by the carbon accumulation in the mixed

layer. Since the carbon accumulation accelerates (following the accelerating increase in atmospheric $pCO_2$), this not only leads to a difference in mean flux (Fig. 9) but also to a difference in trend. Due to the lack of information to correct the 1960–2019 secular trend from the $pCO_2$ data as discussed under (1), this issue leads to an overestimation of the trend in the hybrid estimate compared to OCIM. Most GOBMs (mint green) simulate an even flatter 1960–2019 trend than OCIM.

Looking at the linear trend over the better constrained more recent period 1990–2019 (Fig. 10, bottom), the estimate from

the hybrid mapping becomes more independent from the prior. The $pCO_2$-based hybrid estimates tend to show steeper trends than both OCIM and the GOBM simulations. Most other $pCO_2$ mappings (salmon) estimate the trend to be even more negative than the hybrid mapping. However, given the substantial pentade-to-pentade variations in the global flux (as reflected in the error bars), it is not fully clear how well defined the trend over the 1990–2019 period actually is.

The level of constraint in the trend over the different periods is corroborated by the "zero-prior" mapping not using the

secular trend from OCIM as prior (Fig. 11): Even though the zero-prior explicitly interannual mapping (violet) and the explicitly interannual mapping (green) start from priors with very different secular trends (shown in dark and light grey, respectively), their estimated multi-decadal trends during the recent decades are still very close: In well-constrained regions like the tropical Pacific (bottom panel) they are practically identical, while some deviations occur in poorly constrained regions such as the Indian Ocean, adding up to the slight deviations in the global total flux (top). Only in the early decades where there are

hardly any $pCO_2$ data to constrain the estimates, the two mappings stick to the differing multi-decadal trends (and also to the year-to-year variations) of their respective priors.

As the better-constrained trend over the recent decades (after about 1992) is essentially the same as that in the prior of the explicitly interannual mapping, the flat multi-decadal trend of the zero-prior mapping in the early decades is very unlikely to be true. This illustrates that a prior with the correct secular trend (such as the OCIM result used here) is indeed needed to

extrapolate the ocean $CO_2$ sink into the data-poor first decades of our extended period of interest 1957–2020.





**Data availability**

The sea–air $CO_2$ flux estimates, mapped $pCO_2$ fields, as well as some auxiliary fields are available from the CarboScope website under http://www.bgc-jena.mpg.de/CarboScope/?ID=oc_v2021 where "oc_v2021" can be replaced by the other RunIDs from Table 2. Further results can be made available upon request.

**Author contribution**

C.R. designed and developed the $pCO_2$ mapping algorithm, carried out the estimation runs, and drafted the paper. All other coauthors provided important expertise to interpret the results, reviewed the draft, and gave essential support to finalize the paper.

**Competing interests**

The authors declare that they have no conflict of interest.

*Acknowledgements.* We would like to thank all contributors to the SOCAT data base, which is the basis of this work. We are grateful to J. Orr for kindly providing the code of the *mocsy* package and helping us in its use. C.L.Q. recieved funding from the Royal Society (grant no. RP/R1/191063) and the Natural Environment Research Council Sonata project (NE/P021417/1). T.D. acknowledges support from NSF grant OCE-1948955.

The service charges for this open access publication have been covered by the Max Planck Society.



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



**Table 1.** Input data sets

| Quantity | Data set | Reference | Pre-treatment, original resolution, remarks | Used for |
|---|---|---|---|---|
| $pCO_2$ | SOCATv2021 | Bakker et al. (2016, 2020), http://www.socat.info/ | Data are used having WOCE-flag = 2 and valid fields for fugacity, temperature, and salinity. Values below 200 µatm or above 600 µatm have been excluded as being local compared to the grid cells. Values have been transferred from fugacity to partial pressure by dividing by 0.996. | Main constraint |
| Ocean fraction | Earth2014 | Hirt and Rexer (2015), http://ddfe.curtin.edu.au/models/ Earth2014/data_1min/masks/ MSK2014_landtypes.1min.geod.bin, accessed 2020-11-12 | $1' \times 1'$; using land type 2 ("ocean bathymetry") | Param. |
| SST | Hadley EN.4.2.1 (g10) | Good et al. (2013) | | Param., Expl. var. |
| Ice-free fraction | HadISST 2.2.0.0. | Titchner and Rayner (2014), https://www.metoffice.gov.uk/hadobs/ hadisst2/, accessed 2020-06-05 | $1° \times 1°$, monthly | Param. |
| MLD | LOCEAN | de Boyer Montégut et al. (2004) | $2° \times 2°$, monthly climatology; using "temperature criterion" | Param. |
| $u$ | JRA55-do v1.5.0 | Tsujino et al. (2018) | $0.5625° \times 0.5625°$, 3-hourly | Param., Expl. var. |
| Sea-level press. | JRA55-do v1.5.0 | Tsujino et al. (2018) | $0.5625° \times 0.5625°$, 3-hourly | Param. |
| Atm. $XCO_2$ | Jena CarboScope sEXTALL_v2021 | Rödenbeck et al. (2018b) | $5° \times 3.83°$, daily; atmospheric inversion | Param. |
| Alk | CDIAC | Lee et al. (2006) | $1° \times 1°$, monthly climatology | Param. |
| Salinity | WOA01 | Conkright et al. (2002) | $1° \times 1°$, monthly climatology; via Lee et al. (2006) | Param. |
| $PO_4$, Si | WOA05 | Garcia et al. (2006) | $1° \times 1°$, monthly | Param. |
| Sea–air $CO_2$ flux | OCIM | DeVries (2014, updated) | $2° \times 2°$, monthly; using "total flux", decadally smoothed | Prior |

Glossary: LOCEAN = Laboratoire d'océanographie et du climat: expérimentations et approches numériques; NCEP = National Centers for Environmental Prediction; SOCAT = Surface Ocean $CO_2$ Atlas; WOCE = World Ocean Circulation Experiment; WOA = World Ocean Atlas; Param. = Parameterizations; Expl. var. = Explanatory variable



**Table 2.** Mapping runs used in this study. The main results are given in bold; the other runs are used to assess uncertainty ("uncertainty cases", Sect. 2.1.4), to illustrate specific points of discussion ("test cases", Sect. 2.1.4), or to assess predictive skill ("cross-validation", Sect. 2.2).

| Run | Representation of $f_{\text{int}}$ | Special feature (if any) | CarboScope RunID |
|---|---|---|---|
| Pre-mapping | Eq. (3) | Lin. of C chem. around $p_a\text{CO}_2$ | ocP_v2021 |
| Zero-prior expl. interann. $p\text{CO}_2$ map. | Eq. (1) | | ocZE_v2021 |
| Explicitly interannual $p\text{CO}_2$ mapping | Eq. (2) | | ocE_v2021 |
| Expl. interann. map. (cross-validation) | Eq. (2) | No $p\text{CO}_2$ data 1995–1999 | ocE_CrossVal5yr1995_v2021 |
| **Multi-linear $p\text{CO}_2$ regression** | Eq. (3) | | ocR_v2021 |
| Multi-linear regr. (uncertainty case RegrSSTNOAA) | Eq. (3) | SST from NOAA_ER | ocR_RegrSSTNOAA_v2021 |
| Multi-linear regr. (uncertainty case RegrU2NCEP) | Eq. (3) | $u^2$ from NCEP reanalysis | ocR_RegrU2NCEP_v2021 |
| Multi-linear regr. (uncertainty case RegrAdddSSTdt2) | Eq. (3) | Added $(\text{d}SST/\text{d}t)^2$ regression term | ocR_RegrAdddSSTdt2_v2021 |
| Multi-linear regr. (uncertainty case RegrAddU4) | Eq. (3) | Added $u^4$ regression term | ocR_RegrAddU4_v2021 |
| Multi-linear regr. (uncertainty case RegrAddpaCO2) | Eq. (3) | Added $p_a\text{CO}_2$ regression term | ocR_RegrAddpaCO2_v2021 |
| Multi-linear regr. (uncertainty case RegrLoose) | Eq. (3) | 4-fold a-priori sigma of regr. terms | ocR_RegrLoose_v2021 |
| Multi-linear regr. (uncertainty case RegrShort) | Eq. (3) | 3-fold shorter a-pr. correlation length | ocR_RegrShort_v2021 |
| Multi-linear regr. (uncertainty case RegrNoDecad) | Eq. (3) | No decadal variab. in explanat. var. | ocR_RegrNoDecad_v2021 |
| Multi-linear regr. (uncertainty case MLDq2) | Eq. (3) | Halved mixed-layer depth | ocR_MLDq2_v2021 |
| Multi-linear regr. (uncertainty case MLDx2) | Eq. (3) | Doubled mixed-layer depth | ocR_MLDx2_v2021 |
| Multi-linear regr. (uncertainty case GasexLow) | Eq. (3) | Reduced piston velocity | ocR_GasexLow_v2021 |
| Multi-linear regr. (uncertainty case GasexHigh) | Eq. (3) | Enhanced piston velocity | ocR_GasexHigh_v2021 |
| Multi-linear regr. (uncertainty case GasexU3) | Eq. (3) | Piston velocity prop. to $u^3$ | ocR_GasexU3_v2021 |
| Multi-linear regr. (test case RegrOnlySST) | Eq. (3) | SST regression term only | ocR_RegrOnlySST_v2021 |
| Multi-linear regr. (test case RegrOnlydSSTdt) | Eq. (3) | dSST/d$t$ regression term only | ocR_RegrOnlydSSTdt_v2021 |
| Multi-linear regr. (test case RegrOnlyU2) | Eq. (3) | $u^2$ regression term only | ocR_RegrOnlyU2_v2021 |
| Multi-linear regr. (test case RegrAddChl_98r19) | Eq. (3) | Added Chl-a regression term[a] | ocR_RegrAddChl_98r19_v2021 |
| Multi-linear regr. (test case 98r19) | Eq. (3) | Regr. 1998–2019 only | ocR_98r19_v2021 |
| Multi-linear regr. (test case RegrHeat_85r09) | Eq. (3) | Repl. dSST/d$t$ by the sea–air heat flux[a] | ocR_RegrHeat_85r09_v2021 |
| Multi-linear regr. (test case 85r09) | Eq. (3) | Regr. 1985–2009 only | ocR_85r09_v2021 |
| Multi-linear regr. (test case RegrCurl_88r18) | Eq. (3) | Replacing $u^2$ by windstress curl[a] | ocR_RegrCurl_88r18_v2021 |
| Multi-linear regr. (test case 88r18) | Eq. (3) | Regr. 1988–2018 only | ocR_88r18_v2021 |
| Multi-linear regr. (cross-validation) | Eq. (3) | No $p\text{CO}_2$ data 1985–1989 | ocR_CrossVal5yr1985_v2021 |
| Multi-linear regr. (cross-validation) | Eq. (3) | No $p\text{CO}_2$ data 1990–1994 | ocR_CrossVal5yr1990_v2021 |
| Multi-linear regr. (cross-validation) | Eq. (3) | No $p\text{CO}_2$ data 1995–1999 | ocR_CrossVal5yr1995_v2021 |
| Multi-linear regr. (cross-validation) | Eq. (3) | No $p\text{CO}_2$ data 2000–2004 | ocR_CrossVal5yr2000_v2021 |
| Multi-linear regr. (cross-validation) | Eq. (3) | No $p\text{CO}_2$ data 2005–2009 | ocR_CrossVal5yr2005_v2021 |
| Multi-linear regr. (cross-validation) | Eq. (3) | No $p\text{CO}_2$ data 2010–2014 | ocR_CrossVal5yr2010_v2021 |
| **Hybrid $p\text{CO}_2$ mapping** | Eq. (4) | | oc_v2021 |
| Hybrid mapping (uncertainty case RegrSSTNOAA) | Eq. (4) | SST from NOAA_ER | oc_RegrSSTNOAA_v2021 |
| Hybrid mapping (uncertainty case RegrU2NCEP) | Eq. (4) | $u^2$ from NCEP reanalysis | oc_RegrU2NCEP_v2021 |
| Hybrid mapping (uncertainty case RegrLoose) | Eq. (4) | 4-fold a-priori sigma of regr. terms | oc_RegrLoose_v2021 |
| Hybrid mapping (uncertainty case RegrShort) | Eq. (4) | 3-fold shorter a-pr. correlation length | oc_RegrShort_v2021 |
| Hybrid mapping (uncertainty case RegrNoDecad) | Eq. (4) | No decadal variab. in explanat. var. | oc_RegrNoDecad_v2021 |
| Hybrid mapping (uncertainty case MLDq2) | Eq. (4) | Halved mixed-layer depth | oc_MLDq2_v2021 |
| Hybrid mapping (uncertainty case MLDx2) | Eq. (4) | Doubled mixed-layer depth | oc_MLDx2_v2021 |
| Hybrid mapping (uncertainty case GasexLow) | Eq. (4) | Reduced piston velocity | oc_GasexLow_v2021 |
| Hybrid mapping (uncertainty case GasexHigh) | Eq. (4) | Enhanced piston velocity | oc_GasexHigh_v2021 |
| Hybrid mapping (uncertainty case GasexU3) | Eq. (4) | Piston velocity prop. to $u^3$ | oc_GasexU3_v2021 |
| Hybrid mapping (cross-validation) | Eq. (4) | No $p\text{CO}_2$ data 1995–1999 | oc_CrossVal5yr1995_v2021 |

[a] Regression run only over 1998–2019, 1985–2009, or 1988–2018, respectively



**Table 3.** The components of the contemporary net sea–air $CO_2$ flux ($F_{net}$) according to Hauck et al. (2020), and whether or not they are included in the individual estimates shown in Fig. 8, Fig. 9, and Fig. 10

| Component of $F_{net}$ | Definition | Included in | | | |
| --- | --- | --- | --- | --- | --- |
| | | This study ($F_{net}$) | OCIM "anthr." | GOBMs | Gruber et al. (2019)[2] |
| $F_{ant,ss}$ | $CO_2$ uptake in direct response to the anthropogenic increase in atmospheric $CO_2$ hypothetically under constant pre-industrial circulation | X | X | X | X |
| $F_{ant,ns}$ | modifications of $F_{ant,ss}$ due to anthropogenic climate change and natural climate variability | X | (X)[1] | X | X |
| $F_{nat,ss}$ | steady-state natural fluxes under constant pre-industrial climate conditions; supposed to be zero in the global sum | X | | X | |
| $F_{nat,ns}$ | modifications of $F_{nat,ss}$ due to anthropogenic climate change and natural climate variability | X | (X)[1] | X | |
| $F_{riv,ss}$ | steady-state outgassing of carbon transported into the ocean by rivers, minus the carbon sedimented at the ocean floor | X | | | |
| $F_{riv,ns}$ | modifications of $F_{riv,ss}$ due to climate change and variability as well as anthropogenic land-use changes | X | | | |

[1] for a mean ocean circulation over the industrial era, i.e., the variations are due to temperature and piston velocity variations only; [2] mean 1994–2007 flux only

**Figure 1.** Illustration of the quantities involved in the mixed-layer scheme (time series panels), and the calculations done to connect them (thick-framed boxes). At the arrows on the right of each calculation box, we give its most important environmental input fields (see Table 1). The time series represent the example pixel enclosing the TAO140W mooring location (140° W, 2° N) in the tropical Pacific; they are taken from the results of this study but shown here for illustration only. *Left:* Quantities on the original daily time steps, plotted for 5 example years. *Right:* The same quantities displayed as smoothed yearly averages, which is the way all results will be shown in this paper. The background shading indicates the ENSO phase (Multivariate El Niño Index (MEI) by Wolter and Timlin, 1993).





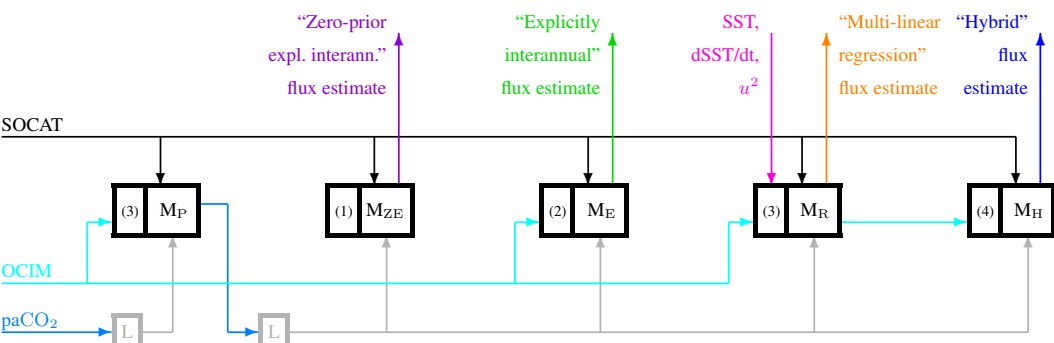

**Figure 2.** Information flow between the presented mapping runs. Thick-framed boxes denote calculations, with arrows denoting their input and output data sets (mostly spatio-temporal fields). The violet, green, orange, and blue arrows represent the spatio-temporal sea–air $CO_2$ fluxes estimated by our main mapping runs $M_{ZE}$, $M_E$, $M_R$, and $M_H$ (these same 4 colors are also used in all line plots in this paper). All mapping runs use the SOCAT data base of $pCO_2$ measurements (point data, black) as their primary information source. The runs are algorithmically identical, except for the representation of the ocean-internal DIC sources/sinks ($f_{int}$) and the corresponding set of adjustable unknowns; this part of the mapping algorithm has therefore been represented explicitly as separate parts of the boxes at their left-hand sides, labelled by the respective equation number. Runs $M_{ZE}$ and $M_E$ are using the representations with explicit interannual degrees of freedom, either without $f_{int}$ prior (Eq. (1)), or using the decadally smoothed OCIM result as $f_{int}$ prior (cyan, Eq. (2)). Run $M_R$ is using the representation involving regression terms (Eq. (3)), which requires the explanatory variables (magenta) as further input fields; it uses the same $f_{int}$ prior as the run $M_E$. The representation of $f_{int}$ in the "hybrid" run $M_H$ again has explicit interannual degrees of freedom, but no long-term and seasonal degrees of freedom (Eq. (4)). Importantly, it uses the ocean-internal DIC sources/sinks estimated by the "multi-linear regression" $M_R$ as its $f_{int}$ prior (cyan again). All mapping calculations use the various input fields shown in Fig. 1 which are however omitted here for clarity.

More technically, the mappings also need a linearization of the non-linear dependence of $pCO_2$ on DIC, consisting of three fields (the derivatives $dpCO_2/dDIC$ as well as reference fields for $pCO_2$ and DIC) together depicted by the grey arrows. Box L denotes the calculation of $dpCO_2/dDIC$ and the reference DIC field from a reference $pCO_2$ field (light blue) as described in Sect. 2.1.6 (the further input fields required by this calculation are not depicted). For the main mappings $M_{ZE}$, $M_E$, $M_R$, and $M_H$, the reference $pCO_2$ field comes from the data-based estimate of the pre-mapping $M_P$. The linearization for $M_P$, in turn, uses the atmospheric $pCO_2$ field as $pCO_2$ reference (light blue again).



**Figure 3.** Yearly sea–air CO$_2$ flux as estimated from $p$CO$_2$ data by the explicitly interannual mapping (green, discarded before 1985 when the data constraint is very weak), the multi-linear regression (orange), and the hybrid mapping (blue). Fluxes have been integrated over a set of regions subdividing the ocean into basins (left to right) and latitude bands (top to bottom). For better temporal orientation across panels, the grey vertical background stripes indicate the positive phases of El Niño-Southern Oscillation according to the Multivariate El Niño Index (MEI) by Wolter and Timlin (1993).



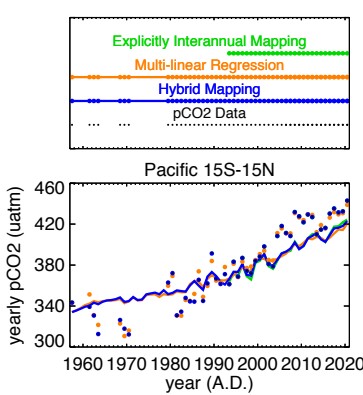

**Figure 4.** Estimated $p\mathrm{CO_2}$ in the tropical Pacific, averaged spatially and over calendar years. The coloured *lines* give full regional averages from the explicitly interannual mapping (green), the multi-linear regression (orange), and the hybrid mapping (blue). The coloured *dots* are from the same estimates, but averaged only over the pixels and time steps covered by $p\mathrm{CO_2}$ data in the respective year. The smaller black dots give the corresponding averages over the data. We note that the green, blue, and black dots are not visible individually because they are almost exactly located on top of each other, indicating that the model–data residuals of the explicitly interannual and hybrid mappings are very small. The differences between dots and lines reflect the bias of the incompletely sampled average compared to the full regional average, which the mapping algorithm is trying to address.





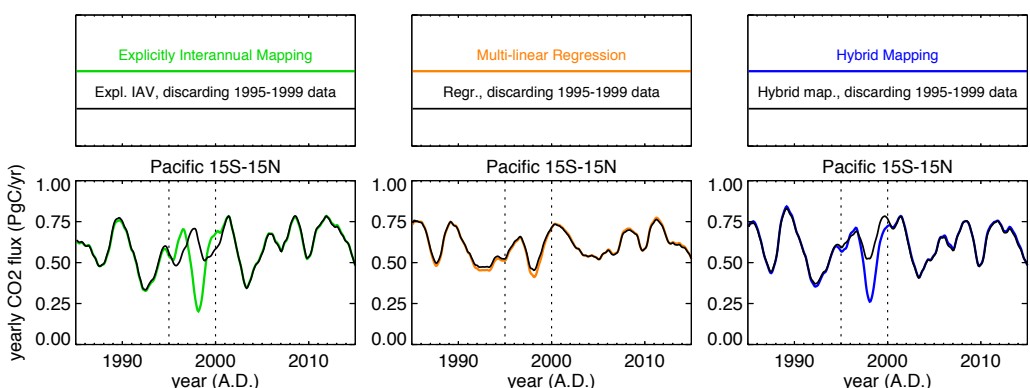

**Figure 5.** Interannual variations of the sea–air $CO_2$ flux in the tropical Pacific estimated by the explicitly interannual mapping (left), the multi-linear regression (middle), and the hybrid mapping (right), either using all $pCO_2$ data (base cases, color) or all data but the ones during 1995–1999 (black)



**Figure 6.** Estimated contributions of the three explanatory variables in the multi-linear regression (as well as the prior, plotted here without its mean) to the ocean internal DIC flux (left) and sea–air $CO_2$ flux (right) in 5 latitudinal bands (top to bottom). Curves show interannual variations as in Fig. 3, the background shading indicates the El Niño phase. In contrast to Fig. 3, fluxes are given in per-area units here, to strengthen the local process perspective; this shifts the relative amplitudes of variability between the regions and allows to use the same y axis in all panels.

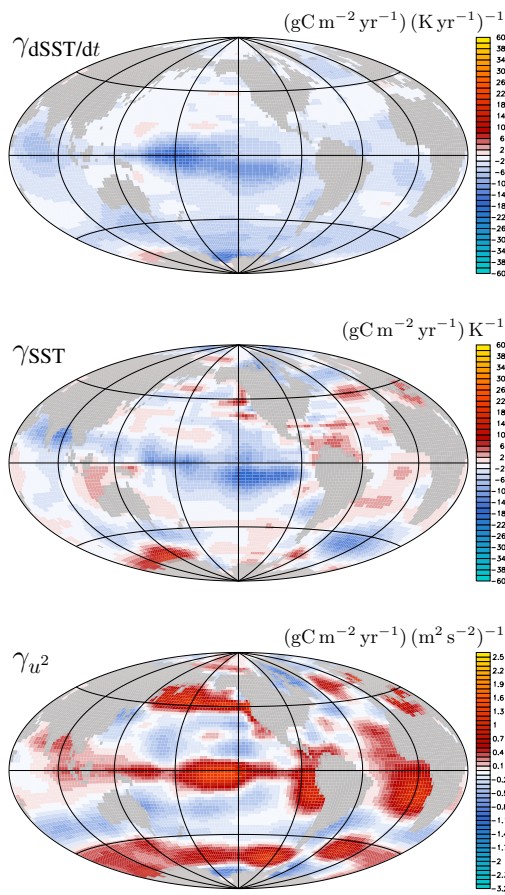

**Figure 7.** Estimated sensitivities of the ocean-internal DIC flux $f_{int}$ against interannual variations in the temporal changes in sea surface temperature (top), in the sea surface temperature itself (middle), and in squared wind speed (bottom). Positive (negative) sensitivities mean that increases in the respective explanatory variable are associated with a stronger source (stronger sink) of DIC in the mixed layer.



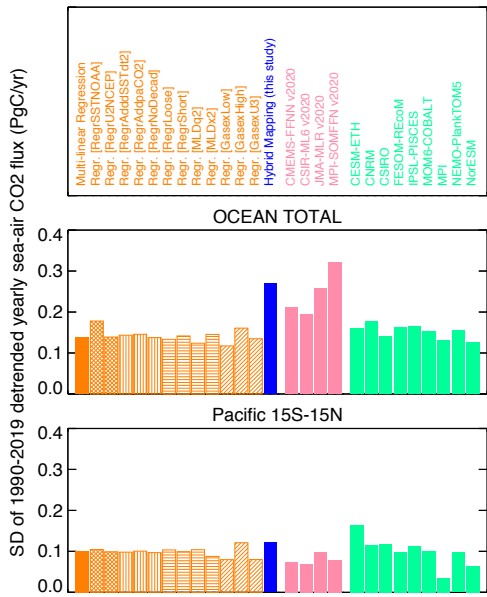

**Figure 8.** Amplitudes of variability of the sea–air $CO_2$ flux on year-to-year time scales around its secular trend, from the multi-linear regression (orange group of bars; solid: base case, hashed: uncertainty cases) and the hybrid mapping (blue) compared to other $pCO_2$ mapping methods (salmon; CMEMS v2020 (Denvil-Sommer et al., 2019), CSIR-ML6 v2020 (Gregor et al., 2019), JMA-MLR v2020 (Iida et al., 2020), and MPI-SOMFFN v2020 (Landschützer et al., 2013)) as well as the ocean biogeochemical process models collated in Friedlingstein et al. (2020) (mint green). The amplitudes are represented by temporal standard deviations of detrended yearly fluxes over the 1990–2019 period. The top panel gives the global flux, the bottom panel the tropical Pacific.





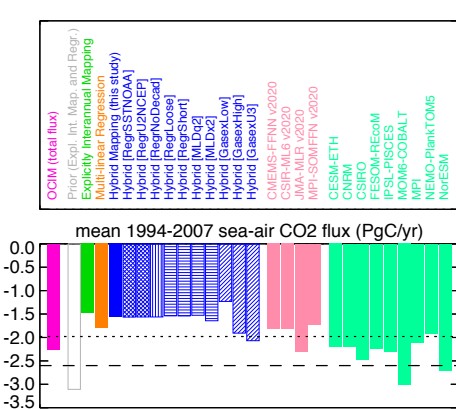

**Figure 9.** Mean global sea–air $CO_2$ flux over 1994–2007 from the hybrid mapping (blue group of bars; solid: base case, hashed: uncertainty cases), compared to other $pCO_2$ mapping methods (salmon; CMEMS v2020 (Denvil-Sommer et al., 2019), CSIR-ML6 v2020 (Gregor et al., 2019), JMA-MLR v2020 (Iida et al., 2020), and MPI-SOMFFN v2020 (Landschützer et al., 2013)), and the ocean biogeochemical process models collated in Friedlingstein et al. (2020) (not including a river-induced sea–air flux [see Table 3], mint green). To the left of the hybrid mapping, we also give OCIM (DeVries, 2014, updated; total contemporary flux) as well as intermediate results from this study. The long-dashed horizontal line indicates the estimate of $-2.6 \pm 0.4\,\mathrm{PgC\,yr^{-1}}$ from ocean interior data by Gruber et al. (2019, anthropogenic carbon only), and the dotted line the same estimate shifted by $0.62\,\mathrm{PgC\,yr^{-1}}$ (mid of Jacobson et al. (2007) and Resplandy et al. (2018)) as an assumed contribution from outgassing of terrestrial carbon transported to the ocean by rivers. Positive fluxes denote oceanic $CO_2$ outgassing into the atmosphere, negative fluxes denote $CO_2$ sinks into the ocean.

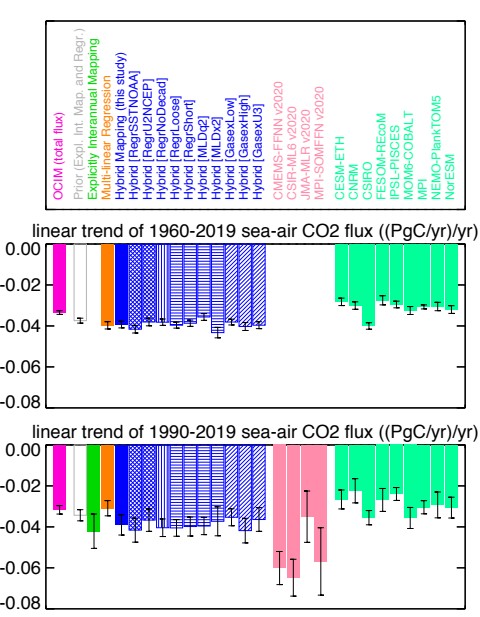

**Figure 10.** Secular linear trend of the global sea–air $CO_2$ flux over 1960–2019 (*top*) and over the more recent 1990–2019 period better constrained by $pCO_2$ data (*bottom*), from the same estimates as in Fig. 9. The error bars give the formal error of the slope calculated from 5-year flux averages, i.e., they reflect the uncertainty due to the pentadal variability around the linear trend, thereby roughly taking into account the serial correlations of the flux e.g. on the El Niño time scale

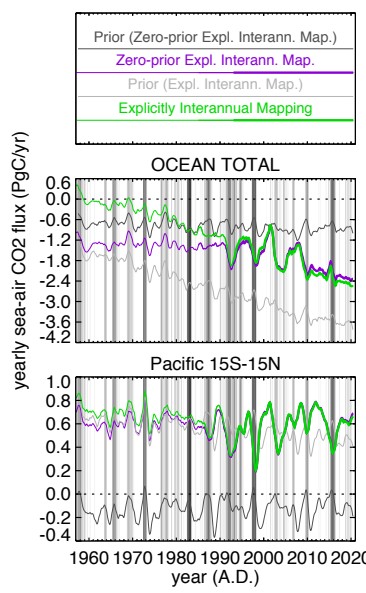

**Figure 11.** Interannually filtered sea–air fluxes as in Fig. 3 (2 example panels only) estimated by the zero-prior explicitly interannual $pCO_2$ mapping (violet) and the explicitly interannual $pCO_2$ mapping (green), as well as their respective priors (dark and light grey, respectively; note that the designation "zero-prior" refers to $f_{int}^{ZE,pri} = 0$, while the a-priori sea–air fluxes shown here are in fact non-zero owing to the rise in atmospheric $CO_2$ and the variations in SST). The line width roughly distinguishes the early period with insufficient data constraint (thin) and the more recent period with better constraint (thick). Vertical background stripes as in Fig. 3