# Peer review of "Data-based estimates of interannual sea-air $CO_2$ flux variations 1957–2020 and their relation to environmental drivers"

_Biogeosciences, 2021_

## Author Response (AR1)

We are giving here a revised version of our answers to the reviewer and community comments posted on the discussion website, reflecting the changes to the manuscript as actually done.

========= Rik Wanninkhof ===================================================

Thank you Rik for reviewing our manuscript and for your interesting comments.

In the following, original comments are quoted in bold italics.

***The authors provide an exhaustive description of the approach and results with a focus on interannual variability but include variability on longer time scales, and compare them with other global air-sea CO2 estimates from other investigations. The document is well referenced and addresses procedural uncertainties very well. The work is of high quality and procedures are meticulously outlined including the assumptions and caveats in the analysis. I see no major shortcomings in the work and my comments are largely based on personal opinion/biases of the various estimates to determine fluxes.***

Thank you for your positive rating.

***- Despite the exhaustive description and illustrative diagrams and figures, the approach to create the flux product is convoluted and remains difficult to understand.***

We checked the methods section for difficult sentences and tried to revise them.

***The step from linear regression to hybrid mapping and***

This step is essentially the addition of an additive correction based on the pCO2 data. We revised this section.

***impact of fint on the results are not completely clear to me.***

On multi-year time scale, the sea-air CO2 flux is essentially identical to the ocean-internal flux f_int. On shorter time scale, the sea-air CO2 flux responds to anomalies in f_int with some time delay, with dampened amplitude, and with modifications on various time scales from variability in sea-air gas exchange.

***- What is the choice of explanatory variables based on? While it is long recognized that SST and MLD are key determinants of pCO2, others are less so***

We indicated the reasons of the choice in Sect 2.1.4.

We agree that MLD would be an attractive variable to be added, but unfortunately it would only be available indirectly from re-analysis of the ocean circulation (i.e., not from observation), which we deem too uncertain (we are aware that the wind speed from atmospheric re-analysis is indirect information as well, which however has been validated against real observations by the meteorological groups).

***- Why is pCO2 used while the primary variable in SOCAT is fCO2? While the conversion between the two is a simple one, the authors chose a constant of 0.996 (Table 1, "Values have been transferred from fugacity to partial pressure by dividing by 0.996."), while the coefficient will differ by about 0.001 (or 0.5 uatm) between 0 and 30 ˚C causing small biases that could be avoided.***

Indeed, the algorithm could also be formulated in terms of fCO2. However, pCO2 is more commonly used in the modelling community, which had influenced the choice during the original implementation in 2013. The bias from using the constant factor 0.996 compared to a more sophisticated SST and SSS dependent factor is very small compared to the many other sources of uncertainty and thus deemed negligible.

***- It is not completely clear what fCO2 data is used. Is it the actual observations, or the gridded product (on monthly basis)? Also it is data sets flagged A and B or all data holding?***

We used the actual observations from file "https://www.ncei.noaa.gov/data/oceans/ncei/ocads/data/0235360/SOCATv2021.tsv" including all obsverations flagged A-D, but not the additional file flagged E.

Thank you for pointing out this omission, we will added this information in the revised manuscript.

***- The authors indicate that the gas transfer velocity (the preferred nomenclature over piston velocity used here)***

We changed the nomenclature in the revised manuscript.

***has a major impact on results and show this, in part, by changing the coefficient of a quadratic dependence and including a cubic***

*dependency. It would be of interest to include a linear dependence as well. See e.g.: Krakauer, N. Y., Randerson, J. T., Primau, F. W., Gruber, N., & Menemenlis, D. (2006). Carbon isotope evidence for the latitudinal distribution and wind speed dependence of the air-sea gas transfer velocity. Tellus B, 58, 390-417, doi: 310.1111/j.1600-0889.2006.00223.x.*

We now ran the suggested test with a linear dependence. It's result differs from the base case (quadratic dependence) approximately in the same absolute way as the cubic dependence but into the opposite direction. We added this case to the revised manuscript (Figs 8-10).

*-While different sets of exploratory variables are used, there is insufficient emphasis that there the quality of exploratory variables before circa 1990 is unknown. This is an added consideration why the results prior to this are not well constrained.*

We agree and added this point more in the revised manuscript (new section 4.6).

*Specific comments:*

*Page 2, lines 7,15, 32 and elsewhere. Reference to " and many others; and many more; and several others " is a bit odd.*

We reconsidered but kept it.

*Page 4, line 15. While not including alkalinity is mentioned later on, it should be stressed the you need Alk and DIC to determine fCO2*

The particular paragraph around Page 4, line 15 is giving an overview of the chain of information implemented, omitting all the details. It seems difficult to mention the Alk influence here without sidetracking from the main thought.

*Page 4. Line 26. Mention size pixels (1 degree by 1 degree by mo???)*

The pixel size is 2.5 degrees by 2 degrees by day (Sect 2.1.7). We added it again in the conclusions.

*Page 6, line 25. See general comment regarding " (2) Observation-based data sets for SST and u are available over all our calculation period 1951–2021 " Prior to 1980- 1990 they are of dubious quality.*

Yes, we added a section on this important issue (new section 4.6).

*Page 9. Line 13. I do not fully understand this sentence "Note that the hybrid run is mathematical equivalent to estimating an additive correction to the multi-linear regression from the pCO2 residuals of the multi-linear regression. "*

This paragraph just responds to past discussions within the more specialized inverse estimation community. It is less relevant to readers mainly interested in the actual results and implications (it only appears in the main text because footnotes are depreciated). We added introductory words to make the role of this paragraph more clear.

*Page 11 line 18. I don't understand this: "even though these decadal trends seem to be the consequence of pronounced anomalies on the faster year-to-year time scale rather than representing actual slowlier decadal variations" Also, "slowlier" is a uncommon word*

The half-sentence tries to express that the primary time scale of variations seems to be about 2-4 years (such as the recurrence time of the ENSO phases). When considering decadal variations (e.g. by decadal smoothing or by calculating trends within 10-year periods), these anomalies on the 2-4 year time scale imply anomalies on decadal time scale, even if they do not actually represent decadal processes. We reformulated this.

*Page 12, line 33 This is an important point: "it indicates that the variability extrapolated into the earlier decades without data will likely be underestimated, too."*

Indeed.

*Page 13 line 24 The following is a bit convoluted, rewrite: "a reduction by 17 to 42% of the increase in the Southern Ocean sink strength (relative to the trend of −0.012(PgC yr−1) yr−1".*

We reformulated.

*Also, is this for the full time period?*

Yes, by "secular trend" we refer to trends over the full period throughout (note that the actual time variation of the wind-speed contribution is not strictly linear). We added the explicit time period.

*Page 14, line 29. "Upwelling both decreases" I'd include "mixing"*

We added "and mixing in from below".

*Page 15, line 3. "Yet, this is still controversially discussed" This is a strange way to end a paragraph.*

The controversy refers to whether the nutrient effect or the carbon effect dominates the upwelling signal (the 2 effects having opposite sign). We reformulated to make that more clear.

*Page 15, line 8. "DIC fluxes" change to "DIC and Alkalinity fluxes"*

Even though changes in pCO2 could indeed be caused by either DIC or Alk fluxes, the suggested formulation would not be appropriate at this particular place ("is estimated to be...") because Alk fluxes are not actually implemented in the estimation.

*Page 16, line 15. This is unclear to me: "For example, in test runs with seasonally resolved (rather than temporally constant) sensitivity coefficients the data could be fitted more closely, but the predictive skill deteriorated (not shown)"*

We noticed that the connection between the 2 sentences was missing, and added it.

*Page 17 line 14: This is a bit unclear. "such that the chlorophyll data may contain substantial variability unrelated to the carbonate system."*

We mean that if variations in the Chl data sets are caused by color variations from processes other than those affecting the carbonate system, then Chl data are not helpful as a predictor in the regression considered here. We reformulated.

*Also, dChl/dt would be an interesting parameter to investigate. However, many have shown the chl is a poor predictor variable for fCO2 so its time derivative probably has little skill as well.*

Indeed, we do not expect more skill from dChl/dt either.

*Page 18 line 20: is the annual average global atm CO2 used? "to atmospheric CO2 (paCO2)."*

We use the decadal average (i.e., decadally smoothed paCO2). We added this missing information.

*Page 19 line 5: typo "internal": "The estimated 5 ocean-interanl DIC"*

Corrected.

*Page 19 line 24: These "afterthoughts" that appear throughout at the end of paragraphs are a little distracting, "of this problem remains for further work."*

We intended to clearly name loose ends.

*Page 21 line 23; minor point Robertson and Watson corrected the pCO2 in water to the lower temperature of the cool skin not the air value, as was done in Woolf et al. "Further, the cooler ocean skin temperature translates the atmospheric pCO2 to a different concentration than that implicitly calculated based on bulk temperature (Robertson and Watson, 1992)."*

We actually meant the DIC concentration in the water.

*Table 1. For completeness add the equation for Fnet: Fnet = Fant,ss +Fant,ns etc, etc.*

We added that.

*Figures: Nice informative figures but they are very "dense" and somewhat difficult to read for the color impaired.*

We were hoping that the most important color contrasts (blue-orange-green in Fig 3 and blue-red in Fig 7) were sufficiently visible to everyone. Other use of color (e.g. in Figs. 2 and 8-10) is more of auxilliary nature but not actually bearing essential information. Please let us know if there are elements that would urgently require revision.

========= Anon. Rev 1 ================================================================

We thank the anonymous reviewer for her/his detailed comments.

In the following, original comments are quoted in bold italics.

***The authors use a well-established method (Rodenbeck et al 2013) to estimate the air-sea CO2 flux from observations and expand it to cover the period 1957-2020 by adding a multi-linear regression approach to form a novel hybrid method. The manuscript is well structured and well written, but the methods (although very well explained) are difficult to grasp. This is a novel approach and I think this study presents a major step forward in estimating the marine CO2 sink.***

Thank you for your positive rating.

For reference below, we would like to point out here that the reviewer's summary is only mentioning 1 out of our 2 equal-ranked topics: We do not only present a spatio-temporal flux product, but also interannual sensitivities ($\gamma_i$) and discuss them in terms of processes (see the title as well as abstract, Introduction, Results, and Conclusion, all mentioning both these topics side by side).

***I do however have one significant issue with the current study and that is the representation of uncertainties. Many recent studies (Bushinsky et al 2019, Gloege et al 2021, Hauck et al 2020, Fay et al 2021, and others) have focused on data limitation and uncertainties, and this should be the standard.***

In addition to our more detailed responses below, we would like to make 2 general remarks about this concern by the reviewer:

First, the reviewer's comment only addresses the hybrid mapping seen as a numerical flux product, even though that is just part of the focus of the study (see above).

Second, an assement of the impact of data density on the 3-dimensional flux field has already been presented for the original CarboScope product in the cited reference Rödenbeck et al. (2014) using the "Reduction of Uncertainty" metric. Regarding the recent decades (SOCAT data period) we do not claim anywhere in the manuscript that the hybrid mapping would be able to improve upon these previous estimates in spatial areas without pCO2 data coverage. Concerning the extrapolation into the early decades, the manuscript clearly states that the interannual variations are likely underestimated and that the secular trend is just coming straight from the OCIM prior.

We added a note into the revised manuscript pointing to the "Reduction of Uncertainty" assement in Rödenbeck et al. (2014).

***While I do believe the authors have done a great job in testing their method using a large variety of sensitivity runs as well as a tough test where 5-year periods are excluded (btw. the same test has been conducted in Landschutzer et al 2016 – supporting information, and should be cited).***

Historically, this test was originally suggested by C.R. in an e-mail to the SOCOM community on 2016-01-08 (back then in a variant "CrossVal5yr0"/"CrossVal5yr1" using alternating 5-year periods with and without pCO2 data). Peter Landschützer kindly did the suggested test runs with the SOM-FFN mapping method. Lacking further participation, the envisaged SOCOM community paper was never written, but Peter Landschützer made at least use of his runs in his 2016 paper. Given this historical background, the citation suggested by the reviewer does not seem appropriate.

***My concerns however are the following:***

***Although sensitivity runs are performed and data omission tests are performed, there is no serious indication of uncertainty of the product. As stated by the authors, one application of this product is to add it to the GCB estimate of the full historical period (page 7 lines 17-20), but how can we have enough confidence in such a reconstruction without thoroughly estimating the uncertainties in the annual mean flux and the presented trends (see also points below)?***

On the one hand (in addition to the general response given above), the study does give uncertainty ranges for GCB-relevant traits (mean, variability, trend) based on the spread across our suite of uncertainty runs (where we do state that this may not comprise the complete uncertainty in the result). On the other hand, the GCB study does not make use of any uncertainty ranges around individual products anyway, but rather does its own uncertainty assessment based on the spread across the ensemble of products from the various groups.

***To provide a more direct example: On page 13 line 22, the authors report a trend of 0.002 – 0.005 PgC/yr/yr – how can they be confident that such a trend is significant?***

The study quantifies a range of the wind-related trend (with the range indicating uncertainties!) and compares it to the total trend. We do not actually make any statements that would depend on the wind-related trend being different from zero. Thus, what type of significance does the reviewer have in mind?

***Furthermore, none of the line plots in the presented figures include error estimates, which causes the impression that there are no uncertainties.***

During manuscript preparation, we had tested presenting the range of uncertainty estimates also in the line plots. The problem, however, is that some of the tests (e.g., GasexLow, GasexHigh) strongly affect the mean flux, thereby shifting the lines vertically and thus disguising any information about the spread in interannual variations anyway (remember that IAV, not the mean, is the actual focus of the study). We had therefore decided to defer the presentation of the range to the bar figures (Figs 8-10) where the effects on

mean and variability are separated from each other.

The new panels on the right in Fig 3 (new counting) now give the results of uncertainty cases (as previously given in the supplement, Fig S4)

***The only exception is figure 10, however, error bars only relate to the linear slope uncertainty and not the method uncertainty.***

The information on uncertainty is provided by the range of results from the various uncertainty cases.

***Furthermore, what is now actually the difference (if any) between this method and the GCB models over the full historical period?***

If "the GCB models" refers to the global ocean biogeochemical models (GOBMs) collated in Friedlingstein et al (2020) and used here for comparison in Figs 8-10 (mint green), there is quite a fundamental difference: GOBMs comprehensively simulate the time evolution of the state of 3D oceanic biogeochemical variables based on natural laws and detailed process parameterizations. The present mapping scheme is primarily driven by the pCO2 observations, making use of some explanatory variables as well as some simple parameterizations of mixed-layer dynamics as needed to relate pCO2 observations and flux fields.

***As it was mentioned in the introduction, I got curious but such an analysis was never presented (this could also serve as validation).***

No, it would not be appropriate to consider GOBM simulations as a validation of observation-based products. Validation can only be made against independent observational data. Unfortunately, this is not an available option here (see the Introduction).

***My biggest concern stems from the lack of historical data (see e.g. Bakker et al 2016 – figure 2). Any estimate before the 1990´s (probaly even before 2000) that is based on SOCAT should be viewed with caution.***

We fully agree that the temporal extrapolation needs to be viewed at with caution. However we feel that this has been mentioned clearly at various places in the manuscript.

***There is no serious attempt here to quantify or at least thoroughly discuss such missing data, maybe with the exception of the Southern Ocean where this is explicitly mentioned. The authors state that the analysis of another method in the Southern Ocean (page 17 lines 3-5) revealed an overestimation of the decadal variability amplitude, but what about this study? Is it in a similar range? At least by adding more regions in Figure 8 one could get an impression.***

See below about assessing the method with synthetic-data tests.

***To provide an example how one could test the results, the authors could use SOCAT data of lower quality flag (assuming that the measurement error may be small compared to the interpolation error),***

Unfortunately, SOCAT data of lower quality flag are far from covering the entire ocean as well. In the data-void ocean areas accounting for much of the interpolation error, such data are thus not available for validation either.

***or by subsampling and reconstructing a hindcast model run (similar to Gloege et al 2021), where a known truth exists.***

Indeed, we have already run tests with synthetic data very analoguous to those presented in the cited study by Gloege et al. (2021). Analysing and presenting these synthetic-data runs, however, needs a separate paper and cannot be added in passing into the present paper.

***Furthermore, a study by Bushinsky et al 2019, using the original version of this method revealed that additional data (in the Southern Ocean) caused a substantial change in the air-sea flux. There is quite some debate about the reliability of the added float data in this study, nevertheless, when considering the historical period, it shows that additional data have the power to substantially change the flux estimate in data sparse regions, which should at least be further discussed.***

We can only repeat that we agree to the concerns of the reviewer, but that we do mention this problem already, while a more quantitative assessment is either impossible due to the lack of independent data or needing a separate paper.

***I am also puzzled by Figure 5 (but this may be a misunderstanding on my side), but does this figure not suggest, that the multi linear regression is more robust in reconstruction periods without any data, whereas the hybrid mapping is not as robust?. Would a multi-linear regression not be more robust then, considering that only a tiny fraction of the ocean is actually covered by observations?***

Indeed there seems to be a misunderstanding. Regression and hybrid mapping are not "alternative methods" we could choose between. Rather, they have different purposes (sensitivities vs. flux variations), and the regression is a part of the hybrid mapping. The discrepancy seen in the right panel of Fig 5 does not indicate a lack of robustness of the the hybrid mapping, but illustrates that the

regression is not able to represent the full amplitude of variations (as discussed in the manuscript).

*I was quite surprised of the authors statement in the main text that chlorophyll did not make a big difference. Figure S7 suggests that chlorophyll makes a substantial difference in the Southern Ocean, maybe in line with Hauck et al 2013?*

As mentioned in the text, we expect the additional spike to be an artifact, not a meaningful signal. By the way, also other pCO2 regression methods in the literature indicate that chlorophyll is not a very helpful explanatory variable.

*While trends and temporal changes are investigated, there is little discussion about spatial features and (again) the uncertainty spatially. A strong focus is set (understandably) on the tropical Pacific Ocean, but there are other regions that are well observed (like the North Atlantic or the North Pacific Oceans) that could serve as a benchmark test how well the method reconstructs the air-sea CO2 flux in space. In the end, the authors present a 3-dimensionsional product (with increased resolution), hence a comparison in space, e.g. with other methods or direct observations from SOCAT or model estimates should be considered to increase the confidence.*

We explicitly state from the beginning (even already in the title) that this is a paper on interannual variability. While spatial variability may indeed be interesting as well, a single paper cannot consider every aspect of a 3D field.

The reviewer is very welcome to analyse spatial signals in the estimated flux field (which is openly available as given in the manuscript). We would be happy to collaborate on that.

*Minor points:*

*.) page 2 lines 14-17: What about ocean inverse estimates that rely on repeat hydrography measurements and an ocean circulation model?*

Indeed, ocean-interior DIC data have been used to estimate sea-air CO2 fluxes. However, while DIC in the ocean interior can constrain the mean flux, it cannot constrain the year-to-tear variability focused on here.

*.) page 2 line 26: SOCAT provides fCO2 not pCO2*

We agree and clarified this in the revised manuscript.

*.) page 3 line 11: I disagree – a response function analysis very well reveals the individual relationships, even in neural networks with many layers and many degrees of freedom*

Of course, by performing additional retrospective analyses on an already-trained neural network, it would be possible to reconstruct input-output relationships. However, linear regression yields this information in the first place (provided the relationships are sufficiently linear).

By the way, we are not aware of any study in the literature that explicitly presented relationships between pCO2 and the drivers of a neural network. For example, Landschützer et al., Science 349, 1221-1224, doi:10.1126/science.aab2620 (2015) rather used parameterized pCO2=pCO2(SST) relationships to discuss process contributions.

*.) page 6 lines 19-21: Salinity may be an important regression variable, particularly in the polar regions*

Thank you for pointing this out. In past tests, we had indeed used Sea Surface Salinity (SSS) as additional explanatory variable. We have re-done such runs now (using SSS and dSSS/dt) and added them into the supplement of the revised manuscript (new Fig S11). They do not change the result much, however. A hesitation about these runs comes from potential measurement problems (e.g., by fouling).

*.) page 7 and following: I am not so sure how much these experiments add to exploring the robustness. In Figure 9 it seems that only the gas exchange experiments make a notable difference when it comes to the mean flux analysis.*

We are surprised by this comment as the reviewer expressed before the opinion that uncertainties had not been sufficiently explored. The various test cases do impact specific aspects of the result, especially the variability. Moreover, even if a given uncertain set-up element turns out to have little impact, we consider this an interesting piece of information, too.

*.) Figure 4: the black dots are difficult to see*

We agree that this is not optimal. However, for consistency with previous papers we had decided to keep blue for the base result and black for data. As the blue and black dots are mostly co-located anyway, there isn't actually any information loss from the poor visibility.

========= Jamie Shutler ===========================================================

Thank you Jamie for your discussion contribution about our study.

In the following, original comments are quoted in bold italics.

***This is an interesting paper and an enjoyable read.***

Thank you for your positive rating of our study.

***on page 21, lines 24 to 29, there is some incorrect information and understanding that has led you to some incorrect conclusions, and these are likley to have had a large impact on the results of the analysis.***

***tThe authors write:***

***Watson et al. (2020) estimated that the sum of these two effects would shift pCO2-based estimates of the mean global CO2 flux by 0.8 to 0.9PgC yr1 (stronger25 sink).***

***which is correct.***

***However the next sentence then says:***

***>So far, however, it is unclear how well the water temperature at the relevant vertical positions can actually be determined (an important source of uncertainty not included in Watson et al. (2020)'s range) and how it varies in space and time.***

***This is sentence is partially correct but also a bit misleading and I the authors may be confusing two different issues. The depth that satelite temperature data are relevant for is well understood and well studied (eg see the informaiton from the international Global High Resolution SST (GHRSST) team and publications eg https://www.ghrsst.org/). What is less clear is how the satelite temperature data align with the top and bottom of the mass bondary layer which is where air-sea gas exchange occurs.***

It seems to us that your explanation does confirm that there are uncertainties about the water temperature as used in your algorithm - which is exactly what we say in the manuscript. Independent of the details, it is thus true that there are larger uncertainties from applying your pCO2 adjustment than suggested by Watson et al. (2020).

While it is clear that sea-air fluxes calculated from pCO2 measurements are affected in some way by the 2 effects under consideration here, there is no consensus in the community so far whether the particular adjustments used in Watson et al. (2020) indeed act to cancel these effects, and what further uncertainty they may add (personal communication with various colleagues).

***The authors then continue to say:***

***In any case, we note that our study mainly considers the variability of the flux, for which the effect of a time-constant correction as in Watson et al. (2020) would cancel out.***

***The authors are confused here as well. The Watson et al work presents two corrections. the first corection focusses on the issue that surface pCO2 data (and their paried temperature data) are all collected at different depths, as sampling depth varies between ships and even within a ship track (eg as the ship changes its ballasting). Whereas the second correction that Watson discuses is the one that the authors can ignore as the authors are interested in variability rather than absolute CO2 sink value.***

***So first correction in Watson et al focusses on re-analysing the SOCAT pCO2 data to common and consistent depth. These methods are published and the data are published each year and the re-analysed version of SOCATv2020 are vailable (Shutler et al 2021) (equivalent data for SOCATv2020 and SOCATv2019 are listed on the SOCAT website). This aspect will be important the work the authors present, as I suspect that some of the variability that the authors characterise in their observation-based data is likely due to the inconsitent and varying depth over which the orignal SOCAT pCO2 data are collected. They authors can easily check for this by repeating their analysis using the re-analsyed and depth consistent SOCAT dataset using the data from the Shutler et al link below. Using the re-analysesed SOCAT data may actually strengthen the conclusions in the paper.***

Even if there may be some spatial and temporal variability in the pCO2 adjustments, the results shown in Fig 1 of Watson et al. (2020) testify that the effect on the estimated fluxes on larger scales is not varying much at all when compared to the signals.

***to help, the issue of how pCO2 data collected at depth is not always representative of the surface water has been recently identifed for Arctic regions by Dong et al 2021. Dong et al show that these issues can result in biased fluxes due to salinity issues. Whereas the Watson work shows that this bias due to temperauter can be more widespread. the theory is well discussed in Woolf et al 2016.***

We note that pCO2 is used in our algorithm not only to calculate gas exchange, but it is also linked to mixed-layer DIC concentration via carbonate chemistry calculation. In this part of the algorithm, the suggested adjustments to the data points would clearly introduce errors.

*Shutler et al (2021) Reanalysed (depth and temperature consistent) surface ocean $CO_2$ atlas (SOCAT) version 2021, https://doi.pangaea.de/10.1594/PANGAEA.939233*

*Dong et al, (2021) Near-Surface Stratification Due to Ice Melt Biases Arctic Air-Sea CO2 Flux Estimates, https://agupubs.onlinelibrary.wiley.com/doi/full/10.1029/2021GL095266*

*Woofl et al 2016 On the calculation of air-sea fluxes of CO2 in the presence of temperature and salinity gradients, https://agupubs.onlinelibrary.wiley.com/doi/full/10.1002/2015JC011427*

========= Val Bennington ==================================================

Thank you Val for your interesting and helpful feedback about our manuscript.

In the following, original comments are quoted in bold italics.

*Data-based estimates of interannual sea–air CO2 flux variations 1957–2020 and their relation to environmental drivers by Christian Rödenbeck, Tim DeVries, Judith Hauck, Corinne Le Quéré, and Ralph F. Keeling*

*The authors reconstruct historical air-sea CO2 fluxes from 1957-2020 using a mixed layer scheme constrained by observations of sea surface pCO2 from the SOCAT database.*

*The importance of SST, interannual variations in SST, and squared wind speed to the internal DIC flux from the mixed layer to depth are investigated using a multi-linear regression.*

*The multi-linear regression technique is used as a prior for a hybrid approach that estimates historical air-sea CO2 exchange from 1957 to 2020.*

*The authors' methodology has clearly been exhaustively researched and many sensitivity studies have been explored to consider alternative options. This is work of a high quality that adds an important and new reconstruction of historical air-sea CO2 fluxes to the scientific field.*

Thank you for your positive overall rating.

*My comments/questions are:*

*The article would be improved if the sections outside of Method could have greater focus on the scientific findings instead of the method.*

Indeed, a large fraction of the manuscript is dealing with methodological issues. However we feel that this material is needed to establish the robustness of the estimates. In the revised version, we tried to make the scientific parts better identifiable by re-ordering of the Results sections.

*For example, much of the discussion I expected to see in terms of comparison to other recent reconstructions was in the Appendix. This is quite interesting, and could be moved to discussion.*

The material in the appendix is limited to the long-term mean and the secular trend of the global flux. The discussion of the long-term mean flux was not placed in the main part because this manuscript is dedicated to variability such that the mean flux is actually off-topic. The discussion of the secular trend would have been a very interesting part of the variability, but unfortunately it turns out that we cannot estimate the secular trend robustly by the presented method.

Concerning comparison to other pCO2 mappings, we feel that further detail would go beyond the scope of a paper like this, presenting a new calculation. We feel that it is better to first introduce the individual products and to only then bring them together in a separate study.

*How do we know whether a "spin up" of 6 years (1951-1957) is adequate?*

The length of the spin-up period was chosen during the development based on test runs with different lengths. The main purpose is to ensure that the initial condition of the mixed-layer budget equation does not influence the solution of the budget equation after the spin-up period any more. This can be tested by direct comparison of the time series after differently long spin-up periods. We added a not into the text.

*Why does including the final year (2021) cause problems?*

We are not sure which statement in the manuscript this question refers to - can you specify?

*Don't we expect the sensitivity of the internal DIC fluxes to SST, wind speed, and interannual SST variations to be smooth because internal DIC fluxes are forced to be smooth within the approach?*

In the regression run (R), the sensitivity fields $\gamma_i$ are directly forced to be smooth by the a-priori correlations implemented. The a-priori correlations of the sensitivity fields have been implemented in the same way as the a-priori correlations of the internal DIC fluxes in the explicitly interannual mapping (E) or the hybrid mapping (H). We revised a possibly misleading formulation in Sect 2.1.1.

We note that the internal DIC fluxes in the regression run are not directly subject to any smoothness constraints, that is, they can become as unsmooth as the explanatory variables are.

*The method is complex. The additional sensitivity studies are challenging to understand as presented in the main text. Moving these fully to supplementary would allow more emphasis on the main results. In the main text, it should be sufficient to state to that the sensitivity tests have been done to explore X,Y,Z, and A,B,C are the primary things learned.*

As mentioned above, we feel that the uncertainty cases are needed to establish the robustness of the estimation. In order to make the Methods section easier anyway, we moved the description of the uncertainty cases to a separate sub-section (new Sect 2.2), in order not to disturb any more the progression from the regression (Sect. 2.1.4) to the hybrid mapping (Sect. 2.1.5).

*Some of the figures have many subplots, which takes away from the readers' ability to see the main findings. A good number of the subplots – those not discussed in the text - could be moved to supplementary to make room for larger subplots of interest.*

We considered how different ocean regions could better be prioritized according to this suggestion. However, we feel that a complete representation of the ocean is actually desirable in Figs. 3 and 6.

*In the comparison in Figure 9 to other estimates, the text acknowledges the uncertainty on the river flux adjustment, but this uncertainty is not presented in the figure. Nor is the uncertainty in Fant from Gruber et al. 2019. If these were included, there would not be far less appearance of discrepancy between estimates. Instead of a dash line and a dotted line, a shaded area would be a better way to present this.*

We agree, thank you for this suggestion. However, after closer consideration, a difficulty is that the uncertainties would have to add up in the dotted line (ie., the uncertainty of the dashed line plus the uncertainty of the difference). As it is an appendix figure only, we left it without this addition.

*It would be helpful to add discussion of what other reconstruction approaches can learn from the findings here.*

We fully understand that colleagues working on pCO2 mapping methods themselves are interested in information helpful to their own developments. However, we feel that explicit statements in this respect are very hard to provide here, because that would necessarily be specific to which particular other technique we would think of. But we are certainly very happy to discuss such questions by direct communication.

*Page 22, line 21-26. Replace colons with periods here.*

Thank you for pointing this out, we changed it.

*Page 22, last paragraph / Figure 10 and 11. Yes, an estimate of the pre-observed trend is needed, but it seems that this method is over-amplifying that trend. Can the authors propose approaches that might improve this going forward?*

We already tested several options, but have not yet found a fully convincing solution. We are working on this issue and hope to improve the trend in a next version.

========= Anon. Rev. 2 ======================================================

We thank the anonymous reviewer for her/his interesting comments.

In the following, original comments are quoted in bold italics.

*The authors reconstructed the sea-air CO2 flux during the period 1957-2020 by applying the mixed-layer scheme method*

*(Rödenbeck et al 2013) in combination with multiple linear regressions between DIC fluxes in the mixed-layer and SST/wind speed. Surprisingly, the reconstruction starts at 1957, which is the dawn of history of surface ocean CO2 observation and seems to have the longest term among similar studies ever done constrained by surface pCO2 observations. Such a study can contribute to performing more accurate atmospheric inversion systems and validating ocean biogeochemical model outputs, as well as evaluating the oceanic CO2 sink evolution during the industrial era. The authors also struggled to analyze ocean biogeochemical processes which dominate the carbon flux in the mixed layer by using their scheme used here. This study provides a long-term reconstruction useful for several communities interested in the global carbon cycles and novel knowledge on ocean biogeochemical process relating to surface ocean CO2. To this end, I think that the manuscript has sufficient values to be published in this journal, after some minor concerns listed below are properly addressed. In addition, major viewpoints of the study had already discussed actively before the time I received the review offer. So, there are little comments I can show here, and I would apologize if some comments were duplicated.*

Thank you for your positive rating.

*I'd like to encourage the authors to improve the study and to revise the manuscript for better understanding*

We revised some formulations to improve the accessibility of the manuscript as given above.

*General comment*

*The authors adopted an approach using the multiple linear regression (MLR) analysis. As they described in the manuscript, MLR has a potential to express oceanographical processes which could alter the carbonate chemistry, though the method is computationally primitive. They used a hybrid method to cover up some demerits of MLR. This succeeded in diminishing the uncertainty of reconstruction, but lacked consistency in the method.*

We are not fully sure what this comment exactly refers to. Indeed, unfortunately, the hybrid method involves some unavoidable temporal inconsistency, as the more data-rich recent decades can be corrected while the data-poor early decades remain identical to the MLR result. Due to this, the variability in the early decades is likely underestimated, as stated in the manuscript. The representation of the carbonate chemistry, however, is consistent for both MLR and hybrid mapping, as both of them just differ in the ocean-internal DIC flux field.

*This study well reconstructed a recent trend of increase in oceanic CO2 sink but failed to give information on the sink evolution in earlier decades because of the use of model output for the period. This might not be a major problem, because the authors mainly focused on the interannual variability of the flux and the mean state of ocean biogeochemical processes, as they said in the manuscript.*

We agree with the reviewer. We would have liked to constrain the secular trend better, but it seems to us that other data than pCO2 needs to be used for that.

*Specific comment*

*P14 3.5.1 to 3.5.3: What is the reason that u2, not dSST/dt and/or SST, has the largest effect on DIC flux in the marginal subtropics and subpolar region, where mixed-layer deepening affects the DIC concentration in the winter? All three are good indicators for mixed-layer development. Is it because mixed-layer deepening is not an interannual but a seasonal phenomenon?*

Indeed, the sensitivities calculated by our MLR represent interannual responses, thus they do not reflect correlations along the seasonal development of the mixed layer.

*P16 L1-4: In general, the wind speed often correlates to SST during the deepening of mixed-layer. It is needed to mention a potential reason why the two are mutually independent.*

We agree that the relative independence between the sensitivities against SST and u2 in the MLR is somewhat surprising. Potentially, there are sufficient differences between the SST and u2 fields in terms of their detailed spatial patterns or sub-monthly temporal changes, which would make their "fingerprints" sufficiently different. We added a note on this.

*P17 L7-15: I was surprised that chlorophyll-a concentrations don't add any information on interannual variability of DIC flux (and I guess that dCHL/dt also cannot add any information too). Once more, is it because blooming is not an interannual but a seasonal phenomenon?*

Maybe chlorophyll-a concentrations would indeed have a stronger impact on seasonal variations, even though the timing and intensity of blooming may also be expected to vary from year to year. Possibly, chlorophyll-a data sets contain too much variability from processes not related to the DIC fluxes, precluding a closer correlation. A weak relationship between chlorophyll-a and pCO2 variability has also been found by other authors.

*P18 L24-26: SST has an obvious secular trend like CO2 concentrations in most of the surface ocean. High-SSTs caused both by long-term global warming and by interannual variability have similar effects on stratification, and the same can be said for the case of wind speed, if any secular trends exist. So, the discussion in 4.3 is reasonable. However, that is not sure in the case that some biological and/or biogeochemical processes influence DIC flux, including changes in biological species, which should be considered when long-term analyses be done. Please consider adding more explanations about that, if needed.*

Thank you for pointing out these processes involving trends. As the method does not offer much constraint on the secular trend, unfortunately, it cannot conclude anything about these processes.

*P19 4.5: There are little processes which can alter the alkalinity especially in seasonal to interannual timescales, but it can change due to biological/biogeochemical transition during long-term analyses. Please consider adding more explanations about that, if needed.*

Thank you, we added a note on the biological influence on alkalinity.

---

## Author Response (AR2)

Dear Jack,

herewith we upload the manuscript with all the changes done as requested.

Thanks a lot for handling our manuscript.

Best regards
-Christian

**p. 9, l. 13: observations**

Corrected.

**p. 16, l.3 and p. 20, l33. Both occasions you use On the other hand without a on the one hand. Please rephrase.**

We replaced by "However" and "At the same time", respectively.

**p. 18, l. 16 involve**

Corrected.

**p. 19 introduce abbreviation GOBM here in main text (it now is explained only in appendix or did I miss it).**

We removed the abbreviation here, and used the same wording as in the caption to Fig 8.